# Identification of p38 MAPK inhibition as a neuroprotective strategy for combinatorial SMA therapy

Maria J Carlini[1], Jorge Espinoza-Derout [2], Meaghan Van Alstyne [1,2], Sarah Tisdale[1,2], Eileen Workman[2], Marina K Triplett[2], Ivan Tattoli [2], Shubhi Yadav[1], Christopher E Henderson[2,3], D Martin Watterson [4] & Livio Pellizzoni [1,2,3 ✉]

## Abstract

**Spinal muscular atrophy (SMA) is a neurodegenerative disease caused by ubiquitous deficiency in the SMN protein. The identification of disease modifiers is key to understanding pathogenic mechanisms and broadening the range of targets for developing SMA therapies that complement SMN upregulation. Here, we report a cell-based screen that identified inhibitors of p38 mitogen-activated protein kinase (p38 MAPK) as suppressors of proliferation defects induced by SMN deficiency in mouse fibroblasts. We further show that SMN deficiency induces p38 MAPK activation and that pharmacological inhibition of this pathway improves motor function in SMA mice through SMN-independent neuroprotective effects. Using a highly optimized p38 MAPK inhibitor (MW150) and combinatorial treatment in SMA mice, we observed synergistic enhancement of the phenotypic benefit induced by either MW150 or an SMN-inducing drug alone. By promoting motor neuron survival, pharmacological inhibition of p38 MAPK synergizes with SMN induction and enables enhanced synaptic rewiring of motor neurons within sensory-motor spinal circuits. These studies identify the p38 MAPK pathway as a therapeutic target and MW150 as a neuroprotective drug for combination therapy in SMA.**

**Keywords** Spinal Muscular Atrophy (SMA); Survival Motor Neuron (SMN); p38 Mitogen-Activated Protein Kinase (p38 MAPK); Neuroprotection; Combination Therapy
**Subject Categories** Musculoskeletal System; Neuroscience

## Introduction

Spinal muscular atrophy (SMA) is an inherited neurodegenerative disease characterized by loss of motor neurons and skeletal muscle atrophy, leading to motor dysfunction, paralysis and eventually death if left untreated (Tisdale and Pellizzoni, 2015; Wirth et al, 2020). SMA is caused by a ubiquitous reduction in the levels of the survival motor neuron (SMN) protein, which results from homozygous loss of the *SMN1* gene and preservation of the paralog *SMN2* gene (Lefebvre et al, 1995). Due to a single nucleotide difference affecting the splicing of exon 7 (Lorson et al, 1999), *SMN2* produces low levels of full-length functional SMN protein that cannot compensate for the loss of *SMN1*, leading to SMA (Lefebvre et al, 1997). Accordingly, the time of onset and the severity of the disease inversely correlate with *SMN2* copy number and SMN protein levels (Tisdale and Pellizzoni, 2015; Wirth et al, 2020).

The SMN protein has critical functions in RNA regulation, including but not limited to the assembly of small nuclear ribonucleoproteins (snRNPs) that carry out pre-mRNA splicing and 3'-end processing of histone mRNAs (Li et al, 2014; Singh et al, 2017), and increasing evidence from preclinical studies in animal models links RNA processing defects to SMA pathology (Gabanella et al, 2007; Zhang et al, 2008; Ruggiu et al, 2012; Lotti et al, 2012; Van Alstyne et al, 2018; Simon et al, 2019; Tisdale et al, 2022; Lauria et al, 2020; Osman et al, 2020; Tisdale et al, 2013; Simon et al, 2017). Furthermore, although selective degeneration of motor neurons and skeletal muscle atrophy are disease hallmarks, an expanding body of evidence directly implicates dysfunction of multiple neuronal and non-neuronal elements of sensory-motor circuits in the etiology of SMA (Mentis et al, 2011; Fletcher et al, 2017; Vukojicic et al, 2019; Simon et al, 2019; Delestrée et al, 2023; Imlach et al, 2012; Shorrock et al, 2018; Rindt et al, 2015; Pagiazitis et al, 2025; Simon et al, 2025).

In recent years, therapeutic approaches that increase SMN expression through antisense oligonucleotides or small molecules that target *SMN2* splicing, as well as SMN replacement by gene therapy, have been approved for the treatment of SMA patients (Finkel et al, 2016; Mercuri et al, 2018; Mendell et al, 2017; Darras et al, 2021; Baranello et al, 2021; Finkel et al, 2017). Despite evidence of clinical efficacy, it is widely appreciated that none of these therapies represent a cure for the disease (Ravi et al, 2021; Mercuri et al, 2020; Groen et al, 2018; Chaytow et al, 2021). The

[1]Department of Neurology, Columbia University, New York, NY 10032, USA. [2]Department of Pathology and Cell Biology, Columbia University, New York, NY 10032, USA. [3]Center for Motor Neuron Biology and Disease, Columbia University, New York, NY 10032, USA. [4]Department of Pharmacology, Northwestern University, Chicago, IL 60611, USA. ✉E-mail: lp2284@cumc.columbia.edu

response to therapy is variable, and a significant proportion of patients show modest improvement, especially when treatment is delayed. Accordingly, incomplete correction of disease symptoms combined with variability in the clinical response to current treatments represents an outstanding challenge in the SMA field.

To address currently unmet needs of SMA patients, the development of combination therapies that can enhance the clinical benefit of SMN-inducing drugs is critical (Chaytow et al, 2021). This calls for new approaches to target cellular factors and molecular pathways that are disrupted downstream of SMN deficiency. However, a current limitation is the availability of disease-relevant targets that could be modulated pharmacologically. This requires increased knowledge of disease mechanisms as well as identification and validation of novel therapeutic approaches independent of SMN induction as viable entry points for subsequent clinical testing. To date, the most clinically advanced efforts are focused on targeting myostatin activation as a means to counteract muscle weakness and atrophy (Servais et al, 2024; Crawford et al, 2024). In contrast, there has been a paucity of approaches targeting neuroprotection, and the only candidate molecule (olesoxime) that advanced to clinical testing failed to demonstrate benefit in SMA patients (Muntoni et al, 2020; Bertini et al, 2017). Of note, none of these approaches have been designed to target mechanistically established molecular defects induced by SMN deficiency in SMA.

Here, we sought to address these outstanding issues through the discovery of novel cellular targets and preclinical validation of disease-modifying pharmacological approaches that are independent of SMN induction and ideally suited for combinatorial treatment of SMA. To do so, we carried out a cell-based screen for chemical modifiers of SMN biology and identified inhibitors of p38 mitogen-activated protein kinase (p38 MAPK) as suppressors of cellular phenotypes induced by SMN deficiency in cultured mammalian cells. Notably, previous studies to date present seemingly conflicting evidence, suggesting that both activation and inhibition of p38 MAPK may be beneficial in SMA (Farooq et al, 2009, 2013; Simon et al, 2019). We provide proof-of-concept for the therapeutic benefit of pharmacological inhibition of this signaling pathway in a mouse model of SMA—either alone or in combination with an SMN-inducing drug—which occurs through neuroprotective mechanisms promoting motor neuron survival without increasing SMN expression. Importantly, these studies employed an isoform-selective p38αMAPK inhibitor (MW150)—an experimental therapeutic in early-stage trials for neurodegenerative and neuropsychiatric disorders optimized for safety and low risk for drug–drug interaction—that has the desired properties for use in combination therapy. Thus, our findings identify p38 MAPK as a disease-relevant cellular target and MW150 as a candidate small molecule for neuroprotection in SMA.

## Results

### A cell-based chemical screen identifies candidate modifiers of SMN biology

We previously established a cell model system that uses proliferation as a phenotypic readout of functional levels of human SMN produced from the SMN2 gene in NIH3T3 fibroblasts genetically engineered for doxycycline (Dox)-regulated, shRNA-mediated knockdown of endogenous mouse Smn (Li et al, 2013). Addition of Dox to NIH3T3-SMN2/Smn$_{RNAi}$ cells induces Smn depletion by RNAi and severe impairment of cell proliferation, which is dependent on low levels of human SMN produced by the SMN2 gene. We also established a cell-based assay in 96-well format for the direct measurement of NIH3T3 cell number using nuclear staining with Hoechst, followed by automated, whole-well imaging (Li et al, 2013). Here, we used this system in a phenotypic screen for chemical compounds that increase the cell number of Smn-deficient NIH3T3 as a means to identify candidate small molecule modulators of SMN biology.

The screening design included pre-treatment of NIH3T3-SMN2/Smn$_{RNAi}$ cells with Dox for 5 days prior to seeding in 96-well plates, addition of chemical compounds, incubation for 5 additional days in the presence of Dox, and readout of cell number (Fig. 1A). Several parameters of the cell-based assay were optimized for the chemical screen (see Materials and Methods) and validated to ensure accuracy and specificity using SMN-C3—a compound that potently promotes SMN2 exon 7 splicing and was further developed into the FDA-approved Risdiplam (Naryshkin et al, 2014; Sheridan, 2021)—as a benchmarking compound. Accordingly, proliferation of Smn-deficient NIH3T3-SMN2/Smn$_{RNAi}$ cells was strongly increased by SMN-C3 relative to vehicle using our cell-based assay (Fig. 1B). Dose-response analysis of SMN-C3 indicated strong activity (>3-fold increase in cell number at the lowest maximal effective concentration) and high potency (EC50 = 210 nM) as well as specificity because no effects were found on the proliferation of Smn-deficient NIH3T3-Smn$_{RNAi}$ cells that do not contain the SMN2 gene (Fig. 1D).

Using the cell-based assay and design described above, we screened 1307 chemical compounds with known biological activity. Dox-treated NIH3T3-SMN2/Smn$_{RNAi}$ cells were incubated with a single compound per well at a concentration of 10 μM in the presence of Dox for 5 days, followed by cell number determination. The screen was carried out in triplicate and hits were defined as compounds increasing cell number beyond the vehicle-treated mean plus 3 standard deviations (Fig. 1C). According to these criteria, the screen yielded 15 compounds as initial hits, four of which—EO1428, CGP7930, clozapine, and dihydroergotamine mesylate (DHE)—were confirmed after subsequent validation including counter-screen assays for specificity and dose-response experiments (Fig. 1C,D, see also Materials and Methods). The validated compounds increased the proliferation of Smn-deficient NIH3T3-SMN2/Smn$_{RNAi}$ cells in a dose-dependent manner across a range of concentrations and had EC50 values of <10 μM (Fig. 1D), whereas they showed progressive toxicity at concentrations in the hundred micromolar range. Dose-response studies were also performed in Dox-treated NIH3T3-Smn$_{RNAi}$ cells for a first insight into potential mechanisms of action, and EO1428 was the only compound found to promote proliferation in these cells that do not contain the human SMN2 gene (Fig. 1D), raising the possibility that the other validated compounds could act by increasing SMN2 expression. Moreover, differences in the decline in normalized cell number at higher compound concentrations likely reflect the effects of toxicity compounded with the greater proliferation rate of SMN2/Smn$_{RNAi}$ cells relative to Smn$_{RNAi}$ cells, as we previously reported (Li et al, 2013).

Lastly, we investigated the effects of the validated compounds on SMN expression in Dox-treated relative to untreated NIH3T3-SMN2/Smn$_{RNAi}$ cells. RT-qPCR experiments showed that none of

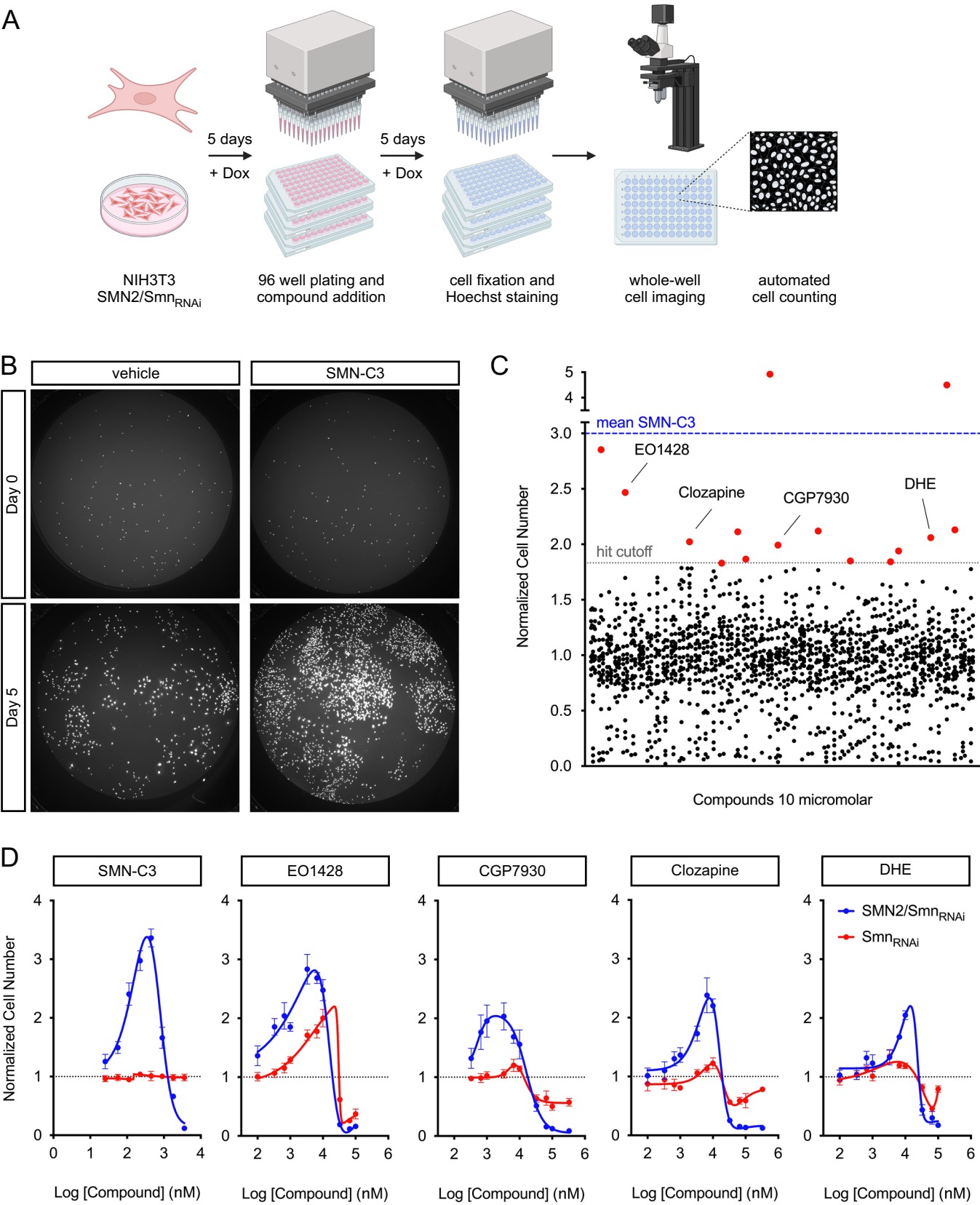

◀   **Figure 1.   A cell-based chemical screen identifies candidate modifiers of SMN biology.**

(A) Schematic of the chemical screen design. NIH3T3-SMN2/Smn$_{RNAi}$ cells were grown for 5 days in the presence of Dox prior to plating in 96-well format plates and addition of compounds from a chemical library. After an additional 5 days in culture in the presence of Dox, cells were fixed and stained with Hoechst, followed by automated whole-well imaging and determination of cell number. Created in BioRender. Pellizzoni, L. (2025) https://BioRender.com/k5m9uz6. (B) Representative whole-well images of Dox-treated NIH3T3-SMN2/Smn$_{RNAi}$ cells cultured in 96-well format in the presence of vehicle (DMSO) or SMN-C3 (500 nM) for 4 h (Day 0) or 5 days (Day 5), followed by fixation and nuclear staining with Hoechst. (C) Graph summarizing the results of the chemical screen. Each point represents the effect of a single compound on the proliferation of NIH3T3-SMN2/Smn$_{RNAi}$ cells at day 5. The mean normalized cell number relative to vehicle-treated cells from three independent biological replicates is shown. The mean of the benchmark compound (SMN-C3) and the hit cutoff are indicated by dotted lines. Primary hits are shown in red, and the four compounds that passed secondary validation are labeled by name. (D) Dose-response analysis of the effects of the indicated compounds on cell proliferation in Dox-treated SMN2/Smn$_{RNAi}$ and Smn$_{RNAi}$ NIH3T3 cells. The graphs show mean and SEM ($n = 6$ independent biological replicates) of normalized cell number relative to vehicle-treated cells at day 5 for each tested concentration, and non-linear curve fitting. Source data are available online for this figure.

the compounds increased the low levels of endogenous mouse Smn mRNA as compared to DMSO treatment (Fig. 2A), indicating that they did not interfere with regulated RNAi in our cell system. Similarly, the compounds did not increase the levels of total and exon 7 containing full-length human SMN mRNA (Fig. 2B,C), excluding direct effects on either transcription or splicing of *SMN2*. Consistent with these findings, Western blot analysis demonstrated that none of the validated compounds increased the low levels of SMN protein in Dox-treated NIH3T3-SMN2/Smn$_{RNAi}$ cells (Fig. 2D,E). Since the compounds act independently of SMN induction, the lower activity of CGP7930, Clozapine, and DHE relative to EO1428 in stimulating proliferation of SMN2/Smn$_{RNAi}$ cells may explain their muted effects in the more severely impaired Smn$_{RNAi}$ cells (Fig. 1D). Regardless of the specific mechanisms, our cell-based screen identified a set of chemical modifiers of cellular proliferation defects induced by SMN deficiency.

## p38 MAPK inhibitors increase the proliferation of SMN-deficient fibroblasts

The identification of EO1428—a small molecule inhibitor of p38 MAPK (Ottosen et al, 2003)—as the top validated hit in our phenotypic screening pointed to inhibition of this pathway as a candidate modifier of SMN deficiency. To investigate this further, we first looked at phosphorylation of Thr180/Tyr182 in p38 MAPK as an established marker of its activation (Plotnikov et al, 2011). Western blot analysis showed a significant increase in the levels of phosphorylated p38 MAPK in Dox-treated, Smn-deficient NIH3T3-SMN2/Smn$_{RNAi}$ cells relative to untreated controls (Fig. 2F,G). Next, we performed dose-response studies of the effects of several chemically distinct p38 MAPK inhibitors, including the highly specific p38α MAPK inhibitors MW108 and MW150 (Roy et al, 2015; Watterson et al, 2013; Roy et al, 2019), on the proliferation of SMN-deficient NIH3T3-SMN2/Smn$_{RNAi}$ cells. Importantly, despite expected differences in potency and activity, all the inhibitors were effective in counteracting SMN-dependent cell proliferation deficits (Fig. 2H). These results indicate that the cell proliferation phenotype induced by SMN deficiency in mouse fibroblasts is mediated, at least in part, by p38 MAPK activation and can be modified through pharmacological inhibition.

## SMN deficiency induces p38 MAPK activation in the spinal cord of SMA mice

To determine whether SMN deficiency induces p38 MAPK activation in a disease-relevant context in vivo, we used the well-established

SMNΔ7 mouse model of SMA that has homozygous knockout of mouse Smn, contains two copies of the human *SMN2* gene, and is homozygous for the SMNΔ7 transgene (Smn$^{-/-}$;SMN2$^{+/+}$;SMNΔ7$^{+/+}$) (Le et al, 2005). Its pattern of motor unit loss closely matches that seen in untreated Type 1 SMA patients (Lee et al, 2025). We performed a longitudinal analysis of p38 MAPK activation in spinal cords isolated from WT and SMA mice at early (P1), mid (P6), and late (P11) symptomatic stages of the disease by Western blot (Fig. 3). This analysis showed a robust increase in the levels of phosphorylated p38 MAPK in the spinal cord of SMA mice relative to controls from P1 to P6 (Fig. 3A–D), while no differences were found at P11 (Fig. 3E,F). Thus, SMN deficiency induces the activation of p38 MAPK at the onset of SMA pathology in a severe mouse model of the disease.

## p38 MAPK inhibition improves motor function of SMA mice through neuroprotective effects independent of SMN induction

To determine whether p38 MAPK activation contributes to SMA pathogenesis, we investigated the effects of its pharmacological inhibition in SMA mice. Most of the commercially available p38 MAPK inhibitors have the potential for off-target effects due to their mixed kinase inhibition activity beyond p38 MAPK, and limited information is available regarding their CNS penetrance. In contrast, MW150 is a potent and highly selective, blood brain barrier-permeable inhibitor of p38αMAPK that has previously been shown to be safe and effective in mouse models of Alzheimer's disease (AD) and other CNS disorders (Robson et al, 2018; Roy et al, 2015, 2019; Rutigliano et al, 2018; Zhou et al, 2017; Watterson et al, 2013; Frazier et al, 2024) and is currently under clinical development. Therefore, we selected MW150 for these in vivo studies.

First, we investigated the exposure levels of MW150 in plasma and brain of neonatal SMA mice following a single intraperitoneal (IP) injection of four escalating drug concentrations (2.5, 5, 10, and 20 mg/kg) at P10. Brain and plasma samples were collected 3 h after injection, followed by analysis of MW150 concentration by LC-MS/MS. Both plasma and brain showed a dose-dependent linear increase in the levels of MW150 that was proportional to the amount of the drug injected in SMA mice (Fig. EV1A,B). Moreover, a relatively constant brain-to-plasma ratio of ~6 was found at all doses tested (Fig. EV1C). These studies confirmed the outstanding brain permeability of MW150 and demonstrated that MW150 drug exposure levels known to effectively inhibit p38αMAPK activity in other experimental models can be reached in the CNS of SMA mice.

Next, we investigated the effects of MW150 on the phenotype of SMA mice, which fail to gain weight or right themselves and

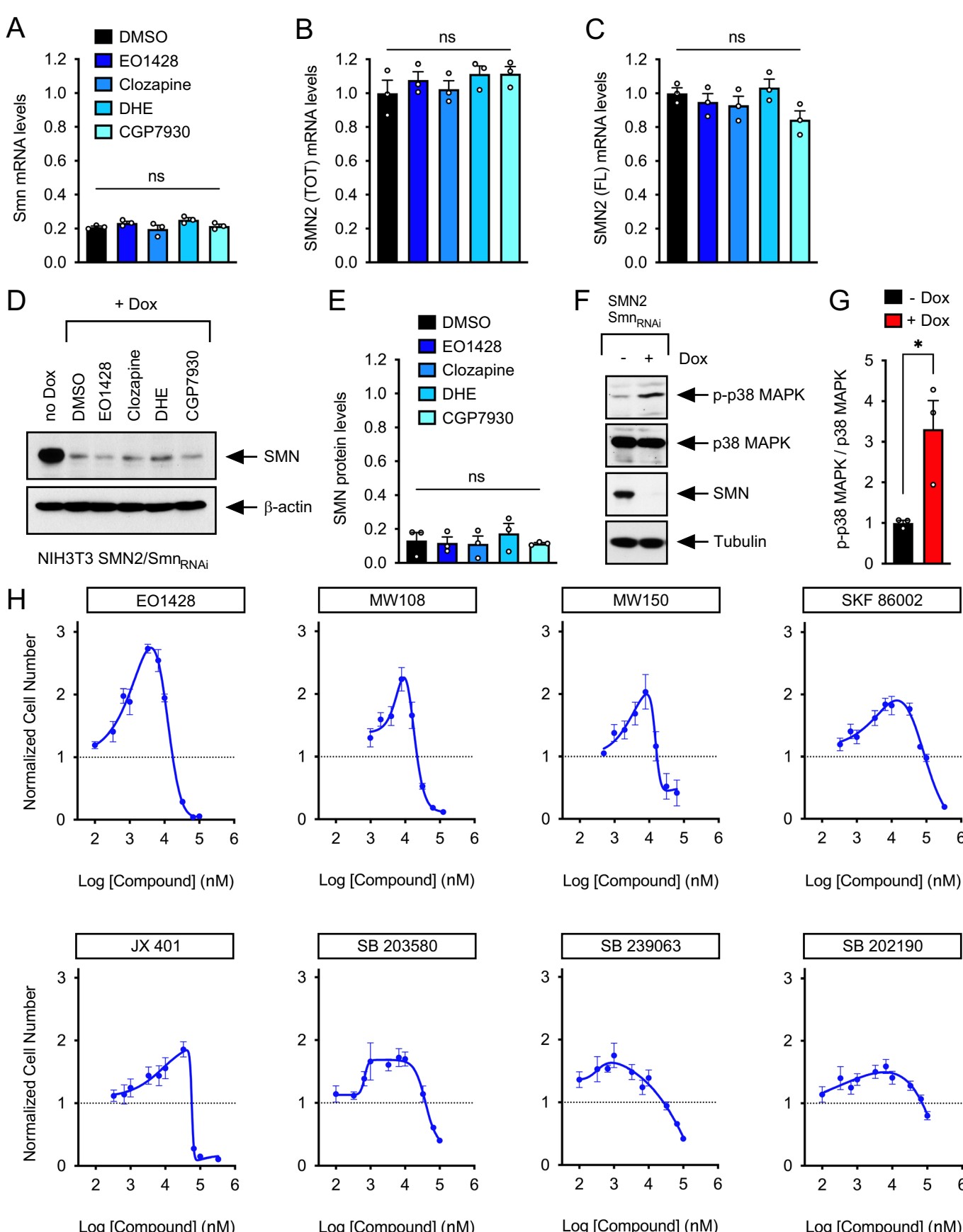

**Figure 2. p38 MAPK inhibition promotes proliferation of SMN-deficient NIH3T3 cells.**

(A) RT-qPCR analysis of *Smn* mRNA levels in untreated and Dox-treated NIH3T3-SMN2/Smn$_{RNAi}$ cells cultured for 5 days in the presence of either DMSO or EO1428, Clozapine, DHE and CGP7930 at a concentration of 10 µM. Mean, SEM, and individual values from Dox-treated cells normalized to DMSO-treated cells cultured without Dox are shown ($n = 3$ independent biological replicates). One-way ANOVA followed by Tukey's post hoc test. ns not significant. (B) RT-qPCR analysis of total (TOT) *SMN2* mRNA levels in the same groups as in (A). Mean, SEM, and individual values from Dox-treated cells normalized to DMSO-treated cells cultured without Dox are shown ($n = 3$ independent biological replicates). One-way ANOVA followed by Tukey's post hoc test. ns, not significant. (C) RT-qPCR analysis of full-length (FL) *SMN2* mRNA levels in the same groups as in (A). Mean, SEM, and individual values from Dox-treated cells normalized to DMSO-treated cells cultured without Dox are shown ($n = 3$ independent biological replicates). One-way ANOVA followed by Tukey's post hoc test. ns not significant. (D) Western blot analysis of SMN expression in the same groups as in (A). β-actin was used as a loading control. (E) Normalized mean, SEM, and individual values of SMN levels in Dox-treated relative to untreated NIH3T3-SMN2/Smn$_{RNAi}$ cells from three independent experiments as in (D). Kruskal–Wallis followed by Dunn's post hoc test. ns not significant. (F) Western blot analysis of total and phosphorylated p38 MAPK in untreated and Dox-treated NIH3T3-SMN2/Smn$_{RNAi}$ cells. SMN and tubulin were used as controls. (G) Quantification of the levels of phosphorylated p38 MAPK relative to total p38 MAPK in NIH3T3-SMN2/Smn$_{RNAi}$ cells cultured with or without Dox from experiments as in (B). Normalized mean, SEM, and individual values from three independent biological replicates are shown. Two-tailed unpaired *t*-test. $P = 0.031$. (H) Dose-response analysis of the effects of the indicated p38 MAPK inhibitors on the proliferation of Dox-treated SMN2/Smn$_{RNAi}$ cells at day 5. The graphs show mean and SEM of normalized cell number relative to vehicle-treated cells at day 5 ($n = 6$ biological replicates) for each tested concentration, and non-linear curve fitting. Source data are available online for this figure.

survive for an average of 14 days postnatally (Le et al, 2005). MW150 was administered to SMA mice beginning at P0 via daily IP injections at a dose of 5 mg/kg, which is the most effective dose established in previous mouse studies (Robson et al, 2018; Roy et al, 2015, 2019; Rutigliano et al, 2018; Zhou et al, 2017; Watterson et al, 2013; Frazier et al, 2024). Importantly, treatment with MW150 improved motor function of SMA mice as measured by the righting time assay relative to vehicle-treated SMA mice (Fig. 4A). The effects on motor behavior were marked but deteriorated together with the overall health of SMA mice when reaching disease end stage prior to death, which is reflected by the higher variability of motor function in the second postnatal week. Accordingly, MW150 treatment had no effect on weight gain or survival of SMA mice (Fig. 4B,C), consistent with MW150 likely improving some but not all of the deficits underlying SMA pathology. We then tested whether MW150 treatment had any effect on the expression of SMN in SMA mice. Western blot analysis did not reveal any effect of MW150 on SMN protein levels in the spinal cord of SMA mice (Fig. 4D,E). Thus, MW150-mediated inhibition of p38 MAPK improves the motor function of SMA mice without inducing SMN.

We previously reported that death of vulnerable motor neurons in SMA mice is driven by the convergence of stabilization and amino-terminal phosphorylation of p53 through distinct mechanisms (Simon et al, 2019, 2017; Van Alstyne et al, 2018), the latter of which is mediated by p38 MAPK activation and inhibited by MW150 (Simon et al, 2019). Therefore, we quantified the total number of lumbar L2 and L5 medial motor column (L5 MMC) SMA motor neurons, which are known to degenerate in the SMNΔ7 model (Mentis et al, 2011; Van Alstyne et al, 2018; Simon et al, 2017). Immunohistochemistry with anti-ChAT antibodies revealed the expected loss of both L2 and L5 MMC motor neurons in the lumbar spinal cord of vehicle-treated SMA mice relative to WT controls at P11 (Fig. 4F–H), while daily treatment with 5 mg/kg of MW150 starting at P0 significantly rescued the survival of these vulnerable motor neuron pools (Fig. 4F–H). Together with our previous study (Simon et al, 2019), these results support the conclusion that the phenotypic benefit of p38 MAPK inhibition on motor function is driven by neuroprotection of SMA motor neurons.

To address this point further, we analyzed the phenotypic effects of daily treatment with MW150 in Smn$^{2B/-}$ mice—a different model of SMA characterized by severe synaptic pathology analogous to that of SMNΔ7 SMA mice (Carlini et al, 2022; Buettner et al, 2021; Eshraghi et al, 2016), but conflicting reports with regard to the extent of motor neuron death. Accordingly, while several studies have reported

significant loss of motor neurons (Eshraghi et al, 2016; Courtney et al, 2019; Reedich et al, 2021; Woschitz et al, 2022), we and others have documented that very little, if any, neurodegeneration occurs in Smn$^{2B/-}$ mice (Carlini et al, 2022; Buettner et al, 2021). Here we found that daily treatment with MW150 had no effects on motor function as well as on weight gain or survival in this mouse model of SMA (Fig. EV2). Smn$^{2B/-}$ mice could not be subjected to combinatorial treatment because they lack *SMN2* genes and are unresponsive to SMN-inducing drugs. Notwithstanding the coarse nature of the motor function tests, these results support our view that motor neuron death is not a prominent driver of SMA pathology in Smn$^{2B/-}$ mice, and that the phenotypic benefit of pharmacological inhibition of p38 MAPK in SMNΔ7 mice results from selective improvement of motor neuron survival.

## An experimental paradigm of combinatorial treatment for evaluation of neuroprotective effects in SMA mice

We next sought to investigate potential synergies of the neuroprotection afforded by MW150 treatment in combination with SMN upregulation. Preclinical studies have shown that SMN-inducing therapies are highly effective in rescuing the disease phenotype in mouse models and do not display the variability in clinical response that is found in the real-world setting of SMA patients. Therefore, we set out to develop an experimental paradigm for suboptimal treatment with SMN-inducing drugs in which to test the neuroprotective effects of p38 MAPK inhibition—an approach often used for combinatorial studies in SMA mice (Feng et al, 2016; Muiños-Bühl et al, 2023; Torres-Benito et al, 2019; Muinos-Bühl et al, 2022; Janzen et al, 2018; Kordala et al, 2023; Dumas et al, 2022). Specifically, we tested the phenotypic response of SMA mice subjected to early or delayed SMN induction by treatment with SMN-C3. The paradigm of delayed treatment was preferred to that of lower doses of SMN-C3 starting from birth to clearly distinguish early, MW150-dependent effects without the confounding element of a concomitant moderate SMN induction.

We performed daily IP injections of SMN-C3 in SMA mice starting at either P0 (early treatment) or P8 (delayed treatment) using the previously established therapeutic dose of 3 mg/kg. Consistent with previous studies (Naryshkin et al, 2014), SMN-C3 early treatment induced a progressive, strong rescue of motor function assessed by the righting time (Fig. 5A), a marked increase in weight gain (Fig. 5B), and extension of survival relative to vehicle-treated SMA mice up to P30 (Fig. 5C), which was set as the

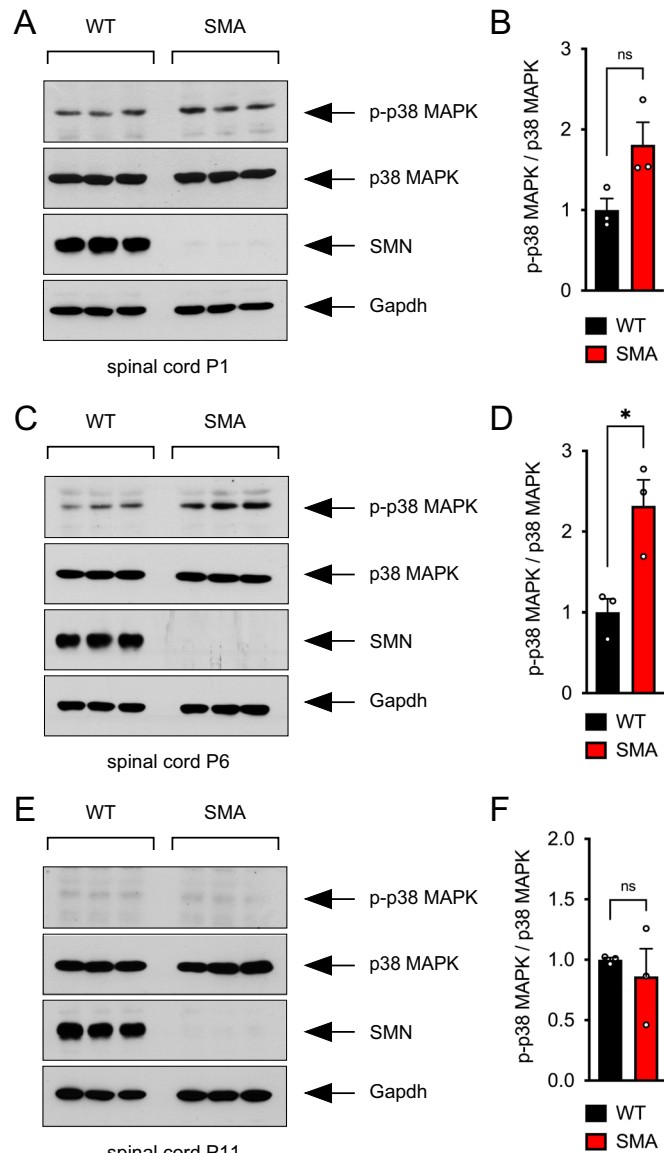

**Figure 3. SMN deficiency induces p38 MAPK activation in the spinal cord of SMA mice.**

(A) Western blot analysis of total and phosphorylated p38 MAPK in the spinal cord of WT and SMA mice at P1. (B) Quantification of the levels of phosphorylated p38 MAPK relative to total p38 MAPK from the experiment in (A). Normalized mean, SEM, and individual values from three independent biological replicates are shown. Mann–Whitney test. ns, not significant. (C) Western blot analysis of total and phosphorylated p38 MAPK in the spinal cord of WT and SMA mice at P6. (D) Quantification of the levels of phosphorylated p38 MAPK relative to total p38 MAPK from the experiment in (C). Normalized mean, SEM, and individual values from three independent biological replicates are shown. Unpaired $t$-test. $P = 0.0224$. (E) Western blot analysis of total and phosphorylated p38 MAPK in the spinal cord of WT and SMA mice at P11. (F) Quantification of the levels of phosphorylated p38 MAPK relative to total p38 MAPK from the experiment in (E). Normalized mean, SEM, and individual values from three independent biological replicates are shown. Welch's $t$-test. ns not significant. SMN and Gapdh were used as controls. Source data are available online for this figure.

time of study termination. In contrast, SMA mice receiving delayed SMN-C3 treatment showed a modest yet variable improvement of motor function towards disease end stage (Fig. 5A), but no significant increase in either weight gain or survival relative to vehicle-treated SMA mice (Fig. 5B,C).

To determine whether SMN-C3 induced comparable increases in SMN expression irrespective of the time of intervention, we analyzed the levels of both exon 7 inclusion in *SMN2* mRNAs and SMN protein in the spinal cord of SMA mice at P11. RT-qPCR analysis showed a similar 2-fold increase in the spinal cord levels of full-length *SMN2* mRNA following either early or delayed delivery of SMN-C3 in SMA mice relative to controls (Fig. 5D). Moreover, SMN protein levels reached about 60% of normal in treated SMA mice regardless of the time SMN-C3 treatment was initiated (Fig. 5E,F). These results indicate that SMN-C3 drug delivery is similarly effective in inducing SMN upregulation irrespective of the time of treatment initiation, and longer treatment does not result in greater SMN induction.

It is conceivable that the observed differences in phenotypic correction reflect accumulation of early damage that cannot be effectively repaired with delayed treatment (Fig. 5A–C), making loss of motor neurons the most likely candidate because it is an irreversible pathogenic event after it has occurred. To address this possibility, we determined the effects of early and delayed treatment with SMN-C3 on the survival of lumbar L2 and L5 MMC motor neurons in SMA mice. Importantly, we found that SMN-C3 treatment at P8 did not prevent motor neuron loss (Fig. 5G–I), which was instead significantly corrected by early treatment. Similar to MW150 (Fig. 4F–H), early treatment with SMN-C3 was more effective in promoting the survival of L5 MMC than L2 SMA motor neurons (Fig. 5G–I), which likely reflects differences in the timing of irreversible commitment to death of these distinct motor neuron pools (Mentis et al, 2011; Kong et al, 2021; Simon et al, 2017).

Together, these results indicate that the lack of neuroprotection correlates with limited phenotypic improvement by delayed SMN induction in SMA mice and that our experimental paradigm of delayed intervention is suitable for evaluating neuroprotective drugs in combination with SMN-inducing therapies.

## Synergistic improvement of the SMA phenotype by MW150 in combination with delayed SMN upregulation

To evaluate whether MW150-dependent neuroprotection provides additional phenotypic benefit relative to delayed SMN induction in SMA mice, we analyzed the phenotypic response of SMA mice that were treated daily with vehicle or MW150 (5 mg/kg) starting at P0 in combination with SMN-C3 (3 mg/kg) treatment at P8 (Fig. 6A). Under these conditions, we found remarkable added benefit of combinatorial treatment with MW150 relative to vehicle (Fig. 6B–D). The improvement in motor function was much more rapid and robust in MW150-treated animals relative to SMA mice treated with delayed SMN-C3 alone (Fig. 6B). We also observed a strong increase in weight gain (Fig. 6C), which either treatment alone fails to induce (compare with Figs. 4B and 5B), and a notable extension of survival in about half of the combinatorially treated SMA mice (Fig. 6D). The strong phenotypic improvement elicited by the

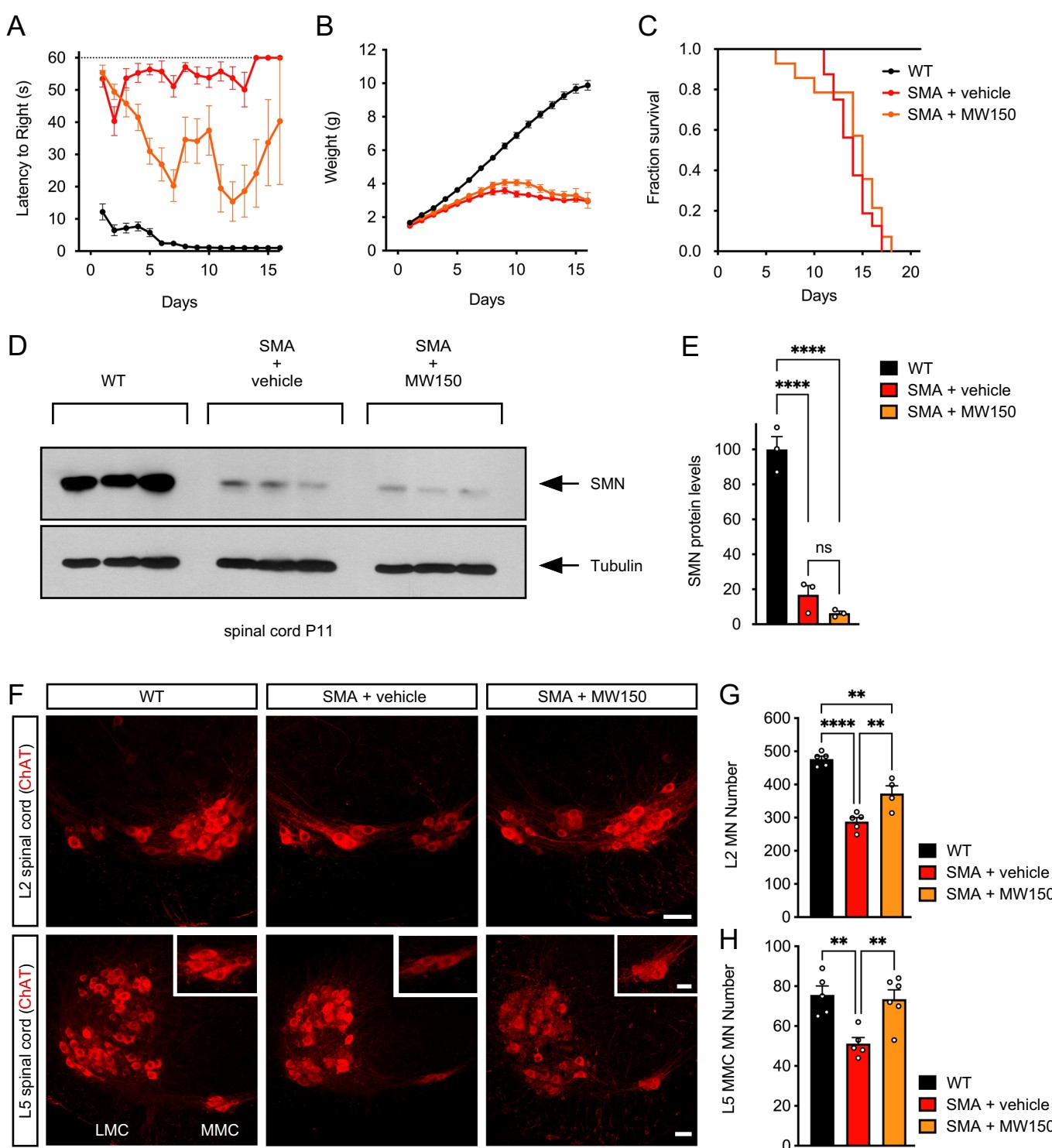

combinatorial treatment was not due to a greater induction in the levels of the SMN protein, which were increased to the same degree in SMN-C3-treated SMA mice with or without MW150 (Fig. EV3). Thus, MW150 treatment strongly enhances the phenotypic benefit of delayed SMN upregulation in SMA mice, highlighting the key impact that preserving motor neuron survival may exert on the potential for correction of SMA pathology by SMN-inducing drugs.

## Neuroprotection by MW150 allows synaptic rewiring of motor neurons by delayed SMN induction

We next sought to establish the basis for the observed synergistic, beneficial effects of MW150 treatment in combination with delayed SMN-C3 treatment. We and others have recently shown that motor neuron deafferentation and neuromuscular junction (NMJ) denervation

◀ **Figure 4. Pharmacological inhibition of p38 MAPK with MW150 improves motor behavior through neuroprotective effects in SMA mice.**

(A) Analysis of righting time in untreated WT mice ($n = 17$) and SMA mice treated daily with vehicle ($n = 16$) or 5 mg/kg MW150 ($n = 14$) starting at P0. Data represent mean and SEM. Mixed-effects model ANOVA comparison of righting time between vehicle and MW150-treated SMA mice: $P < 0.0001$. (B) Analysis of weight gain in the same groups as in (A). Data represent mean and SEM. Mixed-effects model ANOVA comparison of weight gain between vehicle and MW150-treated SMA mice: not significant. (C) Kaplan–Meier analysis of survival in the same groups as in (A). Log-rank (Mantel–Cox) comparison of survival between vehicle and MW150-treated SMA mice: not significant. (D) Western blot analysis of SMN levels in P11 spinal cords from the same experimental groups as in (A-C). Tubulin was used as a loading control. (E) Quantification of SMN levels from the experiment in (D). Normalized mean, SEM, and individual values from three independent biological replicates are shown. One-way ANOVA and Tukey's post hoc test. $P < 0.0001$ (WT vs SMA + vehicle); $P < 0.0001$ (WT vs SMA + MW150); ns not significant (SMA + vehicle vs SMA + MW150). (F) ChAT immunostaining of L2 and L5 spinal cords isolated at P11 from uninjected WT mice and SMA mice treated daily with vehicle or MW150 (5 mg/kg) starting at P0. L5 lateral motor column (LMC) and medial motor column (MMC) motor neuron pools are indicated, and magnified views of L5 MMC motor neurons are shown in the insets. Scale bars = 50 μm and 25 μm (insets). (G) Total number of L2 motor neurons in the same experimental groups as in (F). Normalized mean, SEM, and individual values from the following number of mice: WT ($n = 5$), SMA + vehicle ($n = 5$), and SMA + MW150 ($n = 4$). One-way ANOVA with Tukey's post hoc test. $P < 0.0001$ (WT vs SMA + vehicle); $P = 0.0013$ (WT vs SMA + MW150); $P = 0.0055$ (SMA + vehicle vs SMA + MW150). (H) Total number of L5 MMC motor neurons in the same experimental groups as in (F). Normalized mean, SEM, and individual values from the following number of mice: WT ($n = 5$), SMA + vehicle ($n = 5$), and SMA + MW150 ($n = 6$). One-way ANOVA with Tukey's post hoc test. $P = 0.0046$ (WT vs SMA + vehicle); $P = 0.0064$ (SMA + vehicle vs SMA + MW150). Source data are available online for this figure.

are shared pathogenic features across all SMA mouse models (Buettner et al, 2021; Carlini et al, 2022; Simon et al, 2025). We reasoned that, while delayed treatment with SMN-C3 is ineffective in rescuing vulnerable motor neurons from early commitment to the death pathway (Fig. 5G–I), prior treatment with MW150 rescues these motor neurons (Fig. 4F–H) and could provide the opportunity for SMN-C3 to functionally correct other, p38 MAPK-independent pathological features of SMA. Specifically, we investigated the possibility of synaptic rewiring of motor neurons both centrally—with other neurons of the motor circuit, such as VGluT1+ proprioceptors —and distally—through NMJ reinnervation of skeletal muscle.

We first analyzed neuromuscular junction (NMJ) innervation in the vulnerable, axial muscle quadratus lumborum (QL) of SMA mice at P11 by immunohistochemistry and confocal microscopy. Presynaptic terminals of motor neurons were stained with antibodies against Neurofilament-M and Synaptophysin, while postsynaptic motor endplates were stained with fluorescently labeled bungarotoxin (Fig. 7A). Consistent with previous studies (Tisdale et al, 2022; Simon et al, 2017; Van Alstyne et al, 2018), we found that ~30% of the NMJs are fully denervated in SMA mice at P11 (Fig. 7A,B). Importantly, while this defect was not corrected by treatment with either MW150 alone or delayed SMN-C3, the combinatorial therapy significantly improved NMJ innervation (Fig. 7A,B), albeit not as effectively as early SMN-C3 treatment. Thus, while NMJ denervation in SMA mice is independent from p38 MAPK activation, the neuroprotective effects of MW150 enables early improvement of synaptic connections between motor neuron and muscle by delayed SMN induction.

We then investigated the effects of combinatorial treatment on motor neuron deafferentation. Analysis by immunohistochemistry with ChAT and VGluT1 antibodies followed by confocal microscopy showed that the number of VGluT1+ proprioceptive synapses impinging on the somata of lumbar motor neurons are strongly reduced in SMA mice relative to normal controls at P11 (Fig. 7C,D), consistent with previous studies (Mentis et al, 2011; Fletcher et al, 2017; Simon et al, 2019; Ling et al, 2010). Interestingly, neither delayed treatment of SMA mice with SMN-C3 nor MW150 treatment, either alone or in combination, increased the number of proprioceptive synapses (Fig. 7C,D). In contrast, early SMN-C3 treatment significantly improved motor neuron deafferentation (Fig. 7C,D). These results indicated that the loss of VGluT1+ synapses in SMA mice is independent of p38 MAPK activation and that sensory-motor synaptic connectivity cannot be rapidly reestablished by delayed SMN induction.

To determine whether prolonged SMN induction could further improve synaptic rewiring of SMA motor neurons, we analyzed proprioceptive synapses and NMJ innervation at P21 in SMA mice treated with the combinatorial therapy. WT and early SMN-C3-treated SMA mice were used as controls, while SMA mice treated with vehicle, MW150 alone, and delayed SMN-C3 could not be investigated because they die prior to this time point (Figs. 4C and 5C). Analysis of the number of vulnerable L2 and L5 MMC motor neurons showed no significant differences between WT and SMA mice treated with MW150 and SMN-C3 at P21 (Fig. EV4), consistent with rescue of motor neuron survival by the combinatorial treatment. SMA mice treated early with SMN-C3 alone also showed robust preservation of motor neurons at this later time (Fig. EV4), displaying a significant reduction of L2 motor neurons relative to WT mice that is consistent with the incomplete correction already evident at P11 (compare Figs. 5G,H and EV4A,B). Remarkably, we found no significant differences in the degree of muscle innervation across experimental groups, with essentially all NMJs of the vulnerable QL being fully innervated (Fig. 8A,B). Moreover, the number of VGluT1+ synapses on motor neurons from SMA mice treated with the combinatorial therapy was similar to that of mice treated early with SMN-C3 and not significantly different from WT mice (Fig. 8C,D). Thus, extending SMN-C3 treatment from P11 to P21 not only promoted full NMJ innervation but also the rewiring of proprioceptive synapses on motor neurons, counteracting their early loss (compare Figs. 7D and 8D).

Collectively, these results demonstrate that the neuroprotective effects of MW150 can synergize with SMN-inducing drugs and provide the opportunity to enhance the therapeutic impact of delayed SMN upregulation by promoting synaptic rewiring of SMA motor neurons within spinal sensory-motor circuits. Provided that motor neurons are spared from death, these findings show that even delayed SMN induction can restore functional synaptic connections with proprioceptive neurons and skeletal muscle after their loss, indicating that they are reversible pathogenic events with distinct kinetics for SMN-dependent correction.

## Discussion

The identification of cellular factors and pathways that contribute to pathogenic events downstream of SMN deficiency is key to understand disease mechanisms and broaden the range of targets for developing SMA therapies that can complement SMN

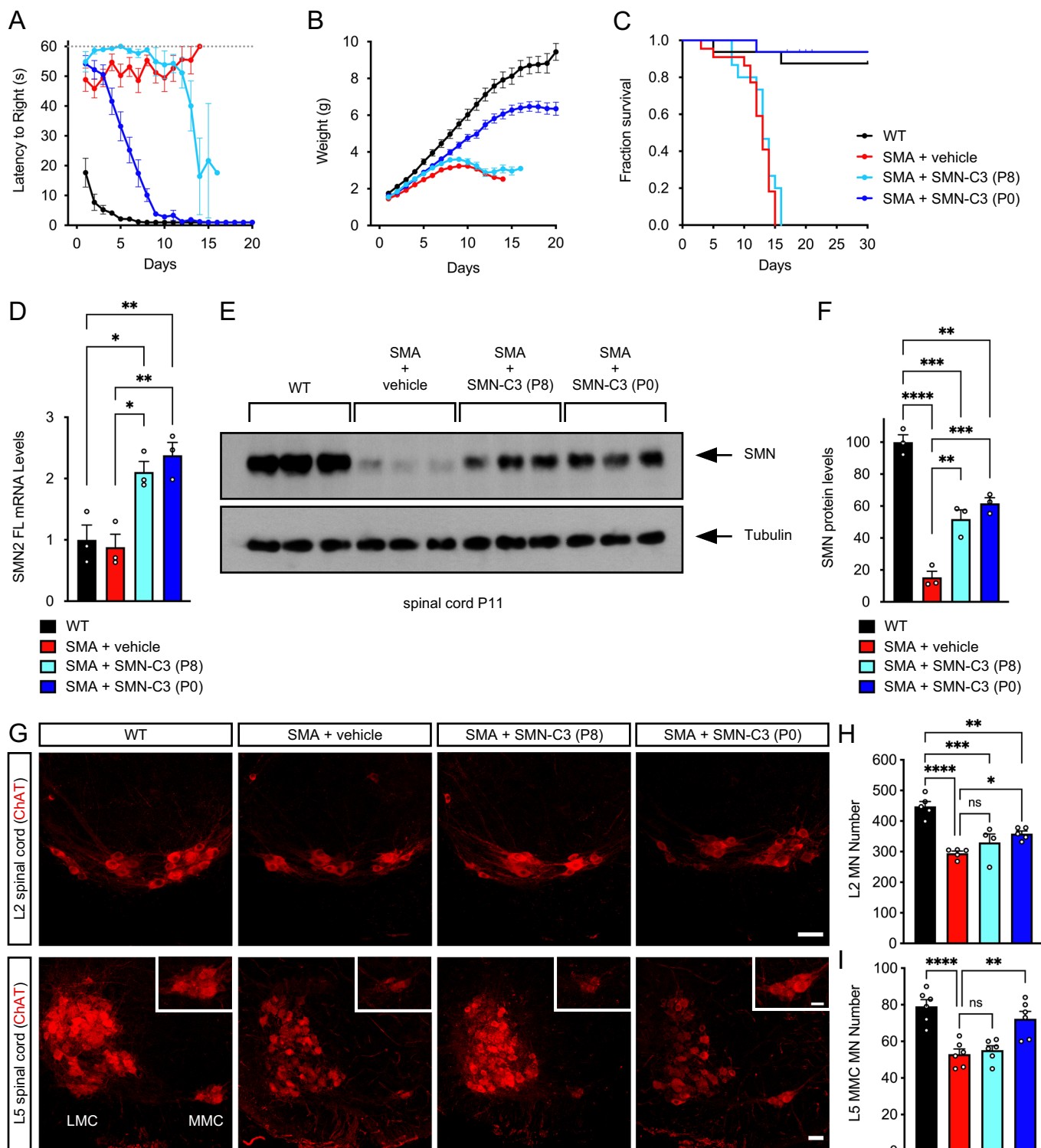

upregulation. Here, we performed a phenotypic screen for chemical modifiers of SMN biology and identified inhibitors of p38 MAPK as suppressors of cell proliferation defects induced by SMN deficiency in mouse fibroblasts. We further show that SMN deficiency induces p38 MAPK activation both in vitro and in vivo, and that pharmacological inhibition of this pathway improves motor

function in SMA mice through neuroprotective effects independent of SMN induction. Using a highly optimized p38αMAPK inhibitor (MW150) and a specific paradigm of combinatorial treatment, we demonstrate synergistic enhancement of the phenotypic benefit induced in SMA mice by MW150 together with an SMN-inducing drug relative to either treatment alone. By promoting survival of

**Figure 5.** Analysis of delayed treatment of SMA mice with SMN-C3.

(A) Analysis of righting time in untreated WT mice ($n = 16$) and SMA mice treated daily with vehicle ($n = 22$) or SMN-C3 (3 mg/kg) starting at P0 ($n = 16$) and P8 ($n = 15$). Data represent mean and SEM. Mixed-effects model ANOVA comparison of righting time: SMA + vehicle vs SMA + SMN-C3 (P8), $P < 0.0001$; SMA+vehicle vs SMA + SMN-C3 (P0), $P < 0.0001$. SMA + SMN-C3 (P8) vs SMA + SMN-C3 (P0), $P < 0.0001$. (B) Analysis of weight gain in the same groups as in (A). Data represent mean and SEM. Mixed-effects model ANOVA comparison of weight gain: SMA + vehicle vs SMA + SMN-C3 (P8), not significant; SMA+vehicle vs SMA + SMN-C3 (P0), $P < 0.0001$; SMA + SMN-C3 (P8) vs SMA + SMN-C3 (P0), $P < 0.0001$. (C) Kaplan–Meier analysis of survival in the same groups as in (A). Log-rank (Mantel–Cox) comparison of survival: SMA + vehicle vs SMA + SMN-C3 (P8), not significant; SMA + vehicle vs SMA + SMN-C3 (P0), $P < 0.0001$; SMA + SMN-C3 (P8) vs SMA + SMN-C3 (P0), $P < 0.0001$. (D) RT-qPCR analysis of the levels of full-length *SMN2* mRNA in P11 spinal cords from the same experimental groups as in (A). Mean, SEM, and individual values normalized to WT samples as a control are shown ($n = 3$ independent biological replicates). One-way ANOVA followed by Tukey's post hoc test. $P = 0.024$ (WT vs SMA + SMN-C3(P8)); $P = 0.0071$ (WT vs SMA + SMN-C3(P0)); $P = 0.014$ (SMA + vehicle vs SMA + SMN-C3(P8)); $P = 0.0043$ (SMA + vehicle vs SMA + SMN-C3(P0)). (E) Western blot analysis of SMN levels in P11 spinal cords from the same experimental groups as in (A–C). Tubulin was used as a loading control. (F) Quantification of SMN levels from the experiment in (E). Normalized mean, SEM, and individual values from three independent biological replicates are shown. One-way ANOVA followed by Tukey's post hoc test. $P < 0.0001$ (WT vs SMA + vehicle); $P = 0.0003$ (WT vs SMA + SMN-C3(P8)); $P = 0.0015$ (WT vs SMA + SMN-C3(P0)); $P = 0.002$ (SMA + vehicle vs SMA + SMN-C3(P8)); $P = 0.0004$ (SMA + vehicle vs SMA + SMN-C3(P0)). (G) ChAT immunostaining of L2 and L5 spinal cords isolated at P11 from uninjected WT mice and SMA mice treated daily with vehicle or SMN-C3 (3 mg/kg) starting at P0 or P8. L5 LMC and MMC motor neuron pools are indicated, and magnified views of L5 MMC motor neurons are shown in the insets. Scale bars = 50 and 25 μm (insets). (H) Total number of L2 motor neurons in the same experimental groups as in (G). Normalized mean, SEM, and individual values from the following number of mice: WT ($n = 5$), SMA + vehicle ($n = 5$), SMA + SMN-C3 (P0) ($n = 5$), and SMA + SMN-C3 (P8) ($n = 4$). One-way ANOVA with Tukey's post hoc test. $P < 0.0001$ (WT vs SMA + vehicle); $P = 0.0005$ (WT vs SMA + SMN-C3(P8)); $P = 0.0038$ (WT vs SMA + SMN-C3(P0)); $P = 0.0391$ (SMA + vehicle vs SMA + SMN-C3(P0)); ns not significant (SMA + vehicle vs SMA + SMN-C3(P8)). (I) Total number of L5 MMC motor neurons in the same experimental groups as in (G). Normalized mean, SEM, and individual values from the following number of mice: WT ($n = 6$), SMA + vehicle ($n = 6$), SMA + SMN-C3 (P0) ($n = 6$), and SMA + SMN-C3 (P8) ($n = 6$). One-way ANOVA with Tukey's post hoc test. $P < 0.0001$ (WT vs SMA + vehicle); $P = 0.0025$ (SMA + vehicle vs SMA + SMN-C3(P0)); ns not significant (SMA + vehicle vs SMA + SMN-C3(P8)). Source data are available online for this figure.

SMA motor neurons, pharmacological inhibition of p38 MAPK synergizes with SMN induction and enables enhanced synaptic rewiring within sensory-motor circuits, which results in increased motor function, weight gain, and survival. Together, our studies identify the p38 MAPK pathway as a therapeutic target and MW150 treatment as a candidate pharmacological approach for neuroprotection with the potential of enhancing the clinical benefit of SMN induction in SMA.

Unbiased phenotypic screens have the power to identify disease modifiers but have not yet been employed as discovery tools to identify disease mechanism-relevant therapeutic targets in SMA. To fill this gap, we performed a cell-based phenotypic screen designed to capture molecules that promote SMN expression or act downstream of SMN depletion. Interestingly, none of the identified hits increase SMN expression, but may function through mechanisms that are independent of SMN induction and, in principle, suitable for use in combination with approved SMA therapies. We prioritized our follow up studies focusing on p38 MAPK because (i) EO1428 was the top validated hit of our screen, (ii) chemically different p38 MAPK inhibitors similarly promoted proliferation of SMN-deficient mouse fibroblasts, and (iii) a highly selective, optimized compound (MW150) was available for in vivo studies (Robson et al, 2018; Roy et al, 2015, 2019; Rutigliano et al, 2018; Zhou et al, 2017; Watterson et al, 2013; Frazier et al, 2024). Furthermore, the p38 MAPK family comprises a group of four serine-threonine protein kinases (α, β, δ, and γ) that process a variety of stimuli and stress signals through phosphorylation cascades and are involved in cell proliferation, differentiation, and death (Cuenda and Rousseau, 2007). Thus, the effects of p38 MAPK inhibitors in promoting cell proliferation of SMN-deficient cells are consistent with the biological role of this pathway. Importantly, accumulation of phosphorylated p38 MAPK in both NIH3T3 cells and the spinal cord of SMA mice provides direct evidence of its activation by SMN deficiency.

To demonstrate the relevance of p38 MAPK in SMA pathology, we investigated the effects of its pharmacological inhibition using MW150. Our findings confirm the outstanding brain permeability of MW150 and show that treatment with this small molecule improves motor function of SMA mice without increasing SMN expression. Mechanistically, p38 MAPK inhibition has neuroprotective effects by acting on the degenerative pathway of SMA motor neurons. We have previously implicated upregulation and phosphorylation of the amino-terminal transcriptional activation domain of the tumor suppressor p53 as two distinct pathogenic events that converge to drive the death of vulnerable motor neuron pools in SMA mice (Simon et al, 2017). We have also shown that both events originate from the disruption of SMN's function in snRNP assembly and downstream dysregulation of specific splicing events (Simon et al, 2019; Van Alstyne et al, 2018). On one hand, alternative splicing changes of specific exons in Mdm2 and Mdm4 pre-mRNAs drive p53 stabilization (Van Alstyne et al, 2018). On the other hand, dysregulated U12 splicing of Stasimon promotes phosphorylation of p53 (Simon et al, 2019). Importantly, the latter involves p38 MAPK activation, whose inhibition by MW150 reduces both p53 phosphorylation and motor neuron death in SMA mice.

Our findings appear to contradict earlier evidence suggesting that p38 MAPK activation may be beneficial in SMA (Farooq et al, 2009, 2013). In a previous study, p38 MAPK activation was shown to increase SMN expression by stabilizing its mRNA in cell models (Farooq et al, 2009). This therapeutic potential was further explored in SMA mice treated with the cyclo-oxygenase 2 inhibitor celecoxib, which activated p38 MAPK and led to a strong induction of SMN in the brain and spinal cord (Farooq et al, 2013). However, despite the robust increase in SMN, the phenotypic improvement in SMA mice was modest. A detailed analysis of motor unit pathology was not conducted in that study, but one possible explanation is that p38 MAPK activation, while increasing SMN, may also have promoted motor neuron death. Notably, our results show that inhibiting p38 MAPK does not reduce SMN expression in vitro or in vivo, effectively ruling out this potentially harmful mechanism.

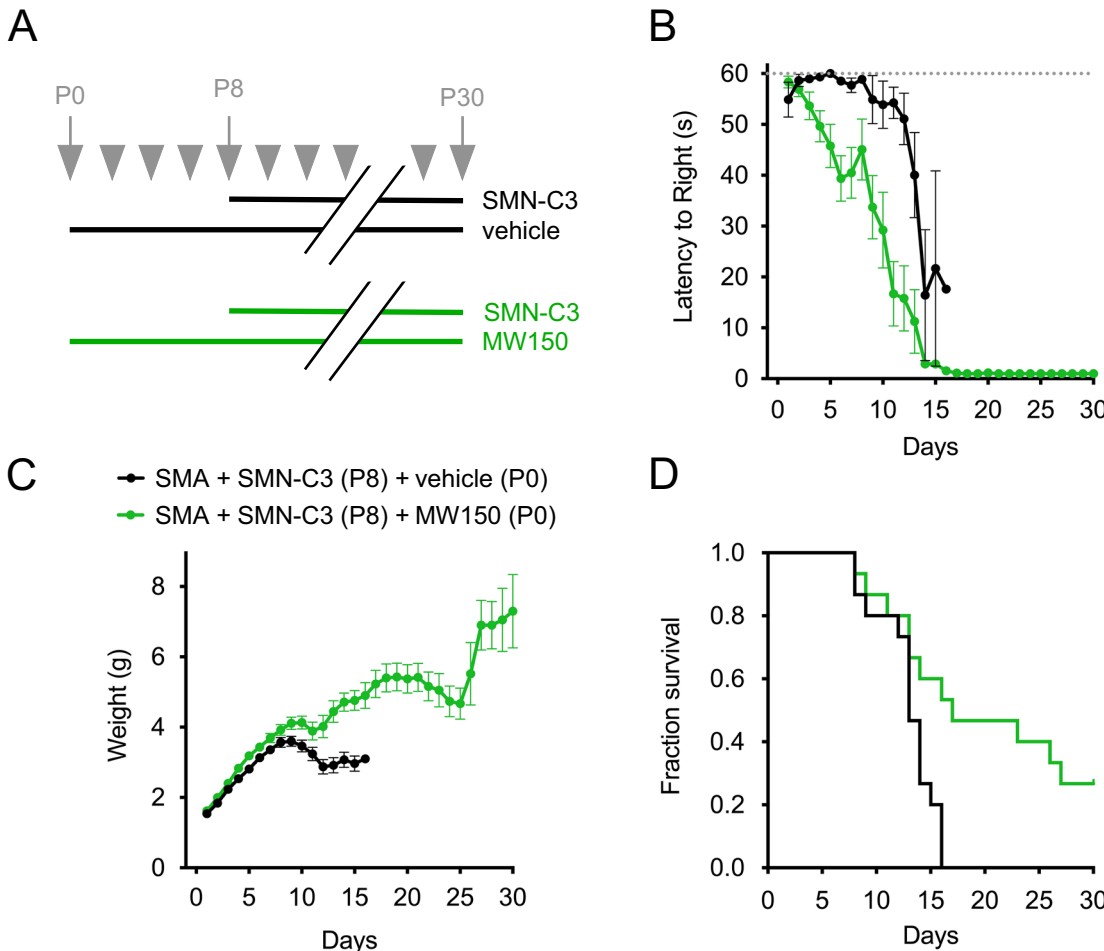

**Figure 6. Phenotypic analysis of combinatorial treatment of SMA mice with MW150 and SMN-C3.**

(**A**) Schematic of the experimental design for delayed SMN-C3 treatment alone and combinatorial treatment with both SMN-C3 and MW150. SMA mice received either daily treatment with vehicle (delayed SMN-C3) or 5 mg/kg MW150 starting at P0 (combinatorial treatment). Daily administration of 3 mg/kg SMN-C3 initiated at P8. (**B**) Analysis of righting time in SMA mice treated daily with delayed SMN-C3 ($n = 15$) or the combinatorial treatment ($n = 15$). Data represent mean and SEM. Mixed-effects model ANOVA comparison of righting time between groups: $P = 0.0002$. (**C**) Analysis of weight gain in the same groups as in (**A**). Data represent mean and SEM. Mixed-effects model ANOVA comparison of weight gain between groups: $P = 0.0003$. (**D**) Kaplan–Meier analysis of survival in the same groups as in (**A**). Log-rank (Mantel–Cox) comparison of survival between groups: $P = 0.0062$. The datasets for the delayed SMN-C3 treatment group are the same as in Fig. 5. Source data are available online for this figure.

Building on the neuroprotective effects of MW150, we explored the therapeutic potential of pharmacological inhibition of the p38 MAPK pathway as a candidate disease-modifying approach for use in combination with SMN-inducing drugs. To do so, we employed an experimental paradigm for suboptimal treatment of SMA mice in which delayed SMN upregulation neither improves motor neuron survival nor synaptic pathology, resulting in the lack of significant phenotypic rescue. Importantly, we document strong synergistic effects when MW150 treatment is used in combination with delayed SMN induction, leading to an improvement of the SMA phenotype that far exceeds that of each treatment alone. The synergy stems from the compounded effects of (i) MW150 promoting motor neuron survival, and (ii) SMN-C3 improving other p38 MAPK-independent SMA deficits. These findings provide direct experimental support for the importance of sparing motor neurons from death in order to achieve the most effective correction of SMA pathology by SMN-inducing drugs. In addition to being a disease hallmark (Tisdale and Pellizzoni,

2015; Wirth et al, 2020), motor neuron loss is an irreversible pathogenic event that cannot be corrected after it has occurred and one that, if insufficiently addressed, can limit the clinical benefit of current SMN-inducing therapies (Finkel et al, 2017; Mercuri et al, 2018; Mendell et al, 2017). In this context, the synergistic effects elicited by pharmacological inhibition of p38 MAPK together with SMN-inducing drugs have clinical implications for combinatorial SMA therapy. However, the combinatorial approach using delayed SMN-C3 treatment was designed to uncover potential synergy with MW150-mediated neuroprotection, while minimizing confounding factors linked to suboptimal therapeutic regimens in mouse models—conditions that do not fully reflect the current clinical setting. Looking ahead to patient translation, MW150 would be administered either concurrently with or prior to SMN-inducing therapies, aiming to enhance motor neuron protection and maximize overall clinical benefit.

Our study provides proof-of-concept that preserving motor neuron survival enables enhanced synaptic rewiring of SMA motor

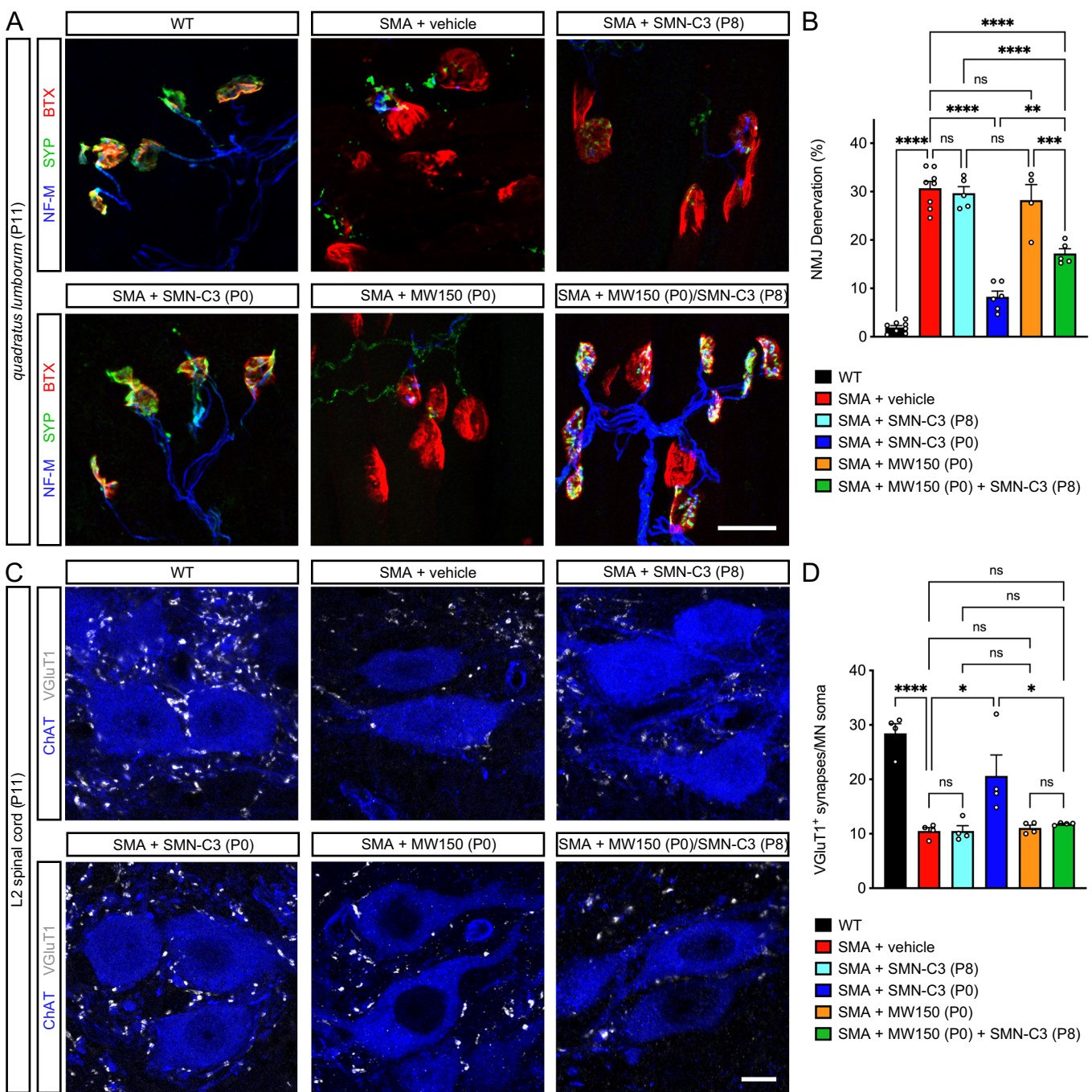

**Figure 7. MW150 treatment promotes partial improvement of NMJs but not central synapses by delayed SMN induction at early times.**

(A) NMJ staining with bungarotoxin (BTX, red), synaptophysin (Syp, green), and neurofilament (NF-M, blue) of QL muscles isolated at P11 from uninjected WT mice and SMA mice treated with vehicle, MW150, early (P0) and delayed (P8) SMN-C3 or with combinatorial therapy. Scale bar = 25 μm. (B) Percentage of fully denervated NMJs in the QL muscle from the same experimental groups as in (A). Normalized mean, SEM, and individual values from the following number of mice: WT ($n = 8$), SMA + vehicle ($n = 8$), SMA + MW150 (P0) ($n = 4$), SMA + SMN-C3 (P0) ($n = 6$), SMA + SMN-C3 (P8) ($n = 5$), and SMA + SMN-C3 (P8)/MW150 (P0) ($n = 5$). One-way ANOVA with Tukey's post hoc test. $P < 0.0001$ (WT vs SMA + vehicle); $P < 0.0001$ (SMA + vehicle vs SMA + SMN-C3(P0)); $P < 0.0001$ (SMA + vehicle vs SMA + MW150(P0) + SMN-C3(P8)); $P < 0.0001$ (SMA + SMN-C3(P8) vs SMA + MW150(P0) + SMN-C3(P8)); $P = 0.0016$ (SMA + SMN-C3(P0) vs SMA + MW150(P0) + SMN-C3(P8)); $P = 0.0004$ (SMA + MW150(P0) vs SMA + MW150(P0) + SMN-C3(P8)); ns not significant. (C) Immunostaining of VGluT1+ synapses (gray) and ChAT+ motor neurons (blue) of L2 spinal cords isolated at P11 from the same experimental groups as in (A). Scale bar = 10 μm. (D) Number of VGluT1+ synapses on the somata of L2 motor neurons from the same experimental groups as in (A). Normalized mean, SEM, and individual values from the following number of mice: WT ($n = 4$), SMA+vehicle ($n = 4$), SMA + MW150 (P0) ($n = 4$), SMA + SMN-C3 (P0) ($n = 4$), SMA + SMN-C3 (P8) ($n = 4$), and SMA + SMN-C3 (P8)/MW150 (P0) ($n = 4$). One-way ANOVA with Tukey's post hoc test. $P < 0.0001$ (WT vs SMA+vehicle); $P = 0.0102$ (SMA + vehicle vs SMA + SMN-C3(P0)); $P = 0.0295$ (SMA + SMN-C3(P0) vs SMA + MW150(P0) + SMN-C3(P8)); ns not significant. Source data are available online for this figure.

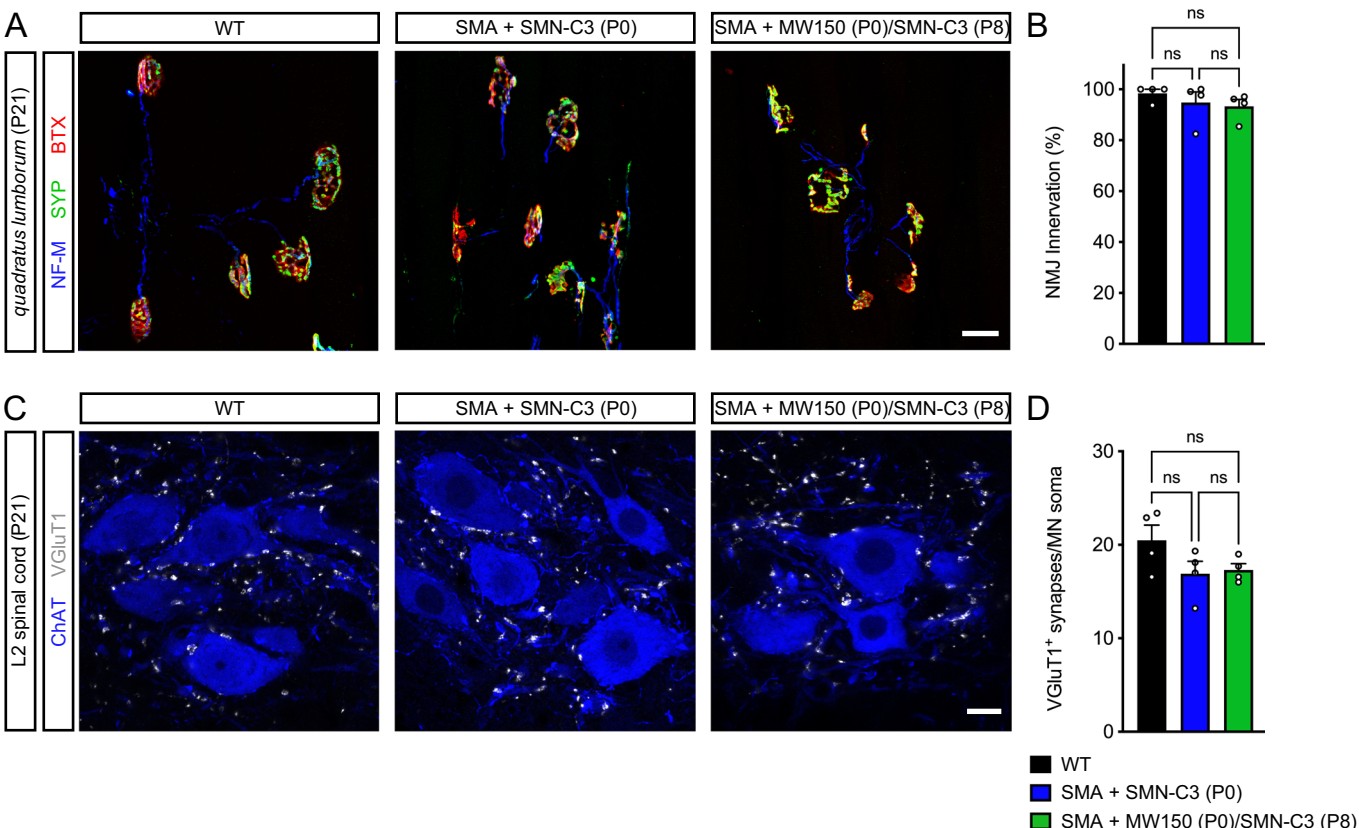

**Figure 8. MW150 treatment promotes restoration of NMJs and central synapses by delayed SMN induction at later times.**

(A) NMJ staining with bungarotoxin (BTX, red), synaptophysin (Syp, green), and neurofilament (NF-M, blue) of QL muscles isolated at P21 from uninjected WT mice, SMA mice injected daily with SMN-C3 (3 mg/kg) starting at P0, and SMA mice injected with SMN-C3 (3 mg/kg) at P8 in combination with MW150 (5 mg/kg) starting at P0. Scale bar = 25 μm. (B) Percentage of fully innervated NMJs in the QL muscle from the same experimental groups as in (A). Normalized mean, SEM, and individual values from the following number of mice: WT (n = 4), SMA + SMN-C3 (P0) (n = 4), and SMA + SMN-C3 (P8)/MW150 (P0) (n = 4). Kruskal–Wallis followed by Dunn's post hoc test. ns not significant. (C) Immunostaining of VGluT1+ synapses (gray) and ChAT+ motor neurons (blue) of L2 spinal cords isolated at P21 from the same experimental groups as in (A). Scale bar = 25 μm. (D) Number of VGluT1+ synapses on the somata of L2 motor neurons from the same experimental groups as in (A). Normalized mean, SEM, and individual values from the following number of mice: WT (n = 4), SMA + SMN-C3 (P0) (n = 4), and SMA + SMN-C3 (P8)/MW150 (P0) (n = 4). One-way ANOVA with Tukey's post hoc test. ns not significant. Source data are available online for this figure.

neurons within spinal sensory-motor circuits and increases phenotypic benefit by SMN upregulation. It also yields insights into the potential for therapeutic correction of synaptic pathology by delayed SMN induction. Although increased NMJ innervation and proprioceptive synaptic inputs on motor neurons have been documented in SMA mice following SMN restoration (Comley et al, 2022; Rimer et al, 2019; Van Alstyne et al, 2021; Simon et al, 2019; Zhao et al, 2016; Feng et al, 2016), these previous studies with early initiation of therapeutic intervention could not clearly distinguish whether SMN induction can act by preventing or slowing down synaptic loss versus reestablishing lost connections. Using our experimental paradigm for combination treatment, we show that delayed SMN induction can promote the formation of new synaptic connections between motor neurons and either proprioceptive neurons or skeletal muscle after they are lost due to SMA pathology. Thus, NMJ denervation and proprioceptive deafferentation are reversible pathogenic events that, unlike motor neuron death, can be recovered even at late symptomatic stages of the disease. Of note, we also show that proprioceptive synapses on motor neurons are reestablished significantly more slowly than

NMJs in SMA mice, revealing that different synaptic connections follow distinct kinetics of restoration after initiation of treatment with SMN-inducing drugs. Collectively, these findings highlight the mechanisms by which the neuroprotective effects of p38 MAPK inhibition can synergize with SMN upregulation to enhance therapeutic efficacy in SMA.

The p38 MAPK pathway has previously been considered a potential therapeutic target for inflammatory diseases and several neurological disorders (Kumar et al, 2003; Chico et al, 2009). Accordingly, p38 MAPK inhibitors have been tested in cellular and animal models of late onset neurodegeneration (Roy et al, 2015; Tortarolo et al, 2003; Xing et al, 2015; Raoul et al, 2002; Watterson et al, 2013; Dewil et al, 2007; Rutigliano et al, 2018; Aikio et al, 2025; Naderi Yeganeh et al, 2024; Dong et al, 2024). Until recently, however, validation of p38 MAPK as a drug discovery target has been limited by the lack of highly selective inhibitors amenable to in vivo use in the CNS (Detka et al, 2024). The development of MW150 has changed this scenario. MW150 is a potent and highly selective inhibitor of p38α MAPK that has previously been shown to be effective in mouse models of AD and

other CNS disorders (Robson et al, 2018; Roy et al, 2015, 2019; Rutigliano et al, 2018; Zhou et al, 2017; Watterson et al, 2013; Frazier et al, 2024). MW150 is a water-soluble and orally bioavailable p38αMAPK inhibitor that has been optimized through medicinal chemistry and exhibits excellent CNS permeability, metabolic stability, safety, and reduced risk for drug–drug interaction. Thus, MW150 has all the desired drug-like properties for the treatment of neurological disorders and is currently under clinical testing. Together with our findings, the favorable pharmacological properties of MW150 should facilitate its entry into the clinical pipeline for testing as a potential combination treatment with SMN-inducing drugs for SMA.

In conclusion, our studies support the therapeutic potential of pharmacologically targeting the p38 MAPK pathway as a candidate neuroprotective approach to enhance motor neuron survival and the clinical benefit of SMN-inducing therapies in SMA.

## Methods

### Reagents and tools table

| Reagent/resource | Reference or source | Identifier or catalog number |
|---|---|---|
| **Experimental models** | | |
| FVB.Cg-Grm7[Tg(SMN2)89Ahmb] Smn1[tm1Msd] Tg(SMN2*delta7)4299Ahmb/J (*M. musculus*) | Jackson Lab | Strain# 005025 |
| FVB.Smn[2B/2B] (*M. musculus*) | PMID: 28172892 | N/A |
| FVB.Smn1[tm1Msd] (*M. musculus*) | PMID: 35913953 | N/A |
| NIH3T3-Smn$_{RNAi}$ | PMID: 23063131 | N/A |
| NIH3T3-SMN2/Smn$_{RNAi}$ | PMID: 23967270 | N/A |
| **Antibodies** | | |
| Rabbit anti-Phospho-p38 MAPK (Thr180/Tyr182) | Cell Signaling | Cat#4511 |
| Rabbit anti-p38 MAPK | Cell Signaling | Cat#8690 |
| Mouse anti-Survival motor neuron (SMN) | BD | Cat#610646 |
| Mouse anti-α-Tubulin | Sigma-Aldrich | Cat#T6199 |
| Mouse anti-beta actin | Proteintech | Cat#66009-1-Ig |
| Mouse anti-glyceraldehyde-3-phosphate dehydrogenase (GAPDH) | Sigma-Aldrich | Cat#MAB374 |
| Peroxidase AffiniPure Goat anti-Mouse | Jackson ImmunoResearch | Cat#115-035-044 |
| Peroxidase AffiniPure Goat anti-Rabbit | Jackson ImmunoResearch | Cat#111-035-003 |
| Goat anti-Choline acetyltransferase (ChAT) | Sigma-Aldrich | Cat#AB144P |
| Guinea pig anti-Vesicular glutamate transporter 1 (VGluT1) | PMID: 28504671 | N/A |
| Guinea pig anti-Synaptophysin1 | Synaptic Systems | Cat#101 308 |
| Rabbit anti-neurofilament-M | Sigma-Aldrich | Cat#AB1987 |
| Cy3 AffiniPure Donkey anti-Goat | Jackson ImmunoResearch | Cat#705-165-147 |
| Alexa Fluor 488 AffiniPure Donkey anti-Rabbit | Jackson ImmunoResearch | Cat#711-545-152 |

| Reagent/resource | Reference or source | Identifier or catalog number |
|---|---|---|
| Cy5 AffiniPure Donkey anti-Guinea Pig | Jackson ImmunoResearch | Cat#706-175-148 |
| **Oligonucleotides and other sequence-based reagents** | | |
| PCR primers | PMID: 22037760 | Table EV1 |
| **Chemicals, enzymes and other reagents** | | |
| Avertin (2,2,2-Tribromoethanol; 2-methyl-2-butanol) | Sigma-Aldrich | Cat#T48402 Cat# 240486 |
| Paraformaldehyde | Electron Microscopy Sciences | Cat#15710 |
| Agar | Fisher Scientific | Cat#A360-500 |
| Normal donkey serum | Millipore | Cat#S30-100ML |
| Fluoromount-G | SouthernBiotech | Cat#0100-01 |
| Optimal cutting temperature (OCT) compound | Scigen/Fisher Scientific | Cat#23-730-625 |
| TRIzol reagent | Thermo Fisher | Cat#15596018 |
| tetramethylrhodamine-conjugated α-bungarotoxin | Invitrogen | Cat#T1175 |
| RevertAid RT Reverse Transcription Kit | Thermo Fisher | Cat#K1691 |
| Ambion DNase I (RNase-free) | Thermo Fisher | Cat#AM2222 |
| *Power* SYBR Green PCR Master Mix | Applied Biosystems | Cat#4367659 |
| RC DC Protein Assay Kit | Bio-Rad | Cat#5000122 |
| Universal Mycoplasma Detection Kit | ATCC | Cat#30-1012 K |
| CELLSTAR µClear 96-well, Flat-Bottom Microplate | Greiner Bio-One | Cat#655090 |
| Tocriscreen Mini | Tocris | Cat#2890 |
| EO1428 | Tocris | Cat#2908 |
| CGP7930 | Tocris | Cat#1513 |
| Clozapine | Tocris | Cat#0444 |
| Dihydroergotamine | Tocris | Cat#0475 |
| SKF 86002 dihydrochloride | Tocris | Cat#2008 |
| JX 401 | Tocris | Cat#2657 |
| SB 203580 | Tocris | Cat#1202 |
| SB 239063 | Tocris | Cat#1962 |
| SB 202190 | Tocris | Cat#1264 |
| MW108 | PMID: 23840427 | N/A |
| MW150 | PMID: 25676389 | N/A |
| SMN-C3 | Combi-Blocks, Inc. | N/A |
| Dimethyl sulfoxide (DMSO) | Sigma-Aldrich | Cat#D2650 |
| Hoechst 33342 | Invitrogen | Cat#H21492 |
| DMEM, high glucose, pyruvate, no glutamine | Gibco | Cat#10313021 |
| HyClone fetal bovine serum (FBS) | Cytiva | Cat#SH30070.01 |
| L-glutamine | Gibco | Cat#25030164 |
| Gentamicin | Gibco | Cat#15750060 |
| Doxycycline HCl | Fisher Scientific | Cat#BP2653-5 |

| Reagent/resource | Reference or source | Identifier or catalog number |
|---|---|---|
| **Software** | | |
| TINA V4.93 | TROPHOS | N/A |
| LAS AF v2.5.2.6939 | Leica | SCR_013673 |
| Canvas X Draw v7.0.4 | Canvas GFX | SCR_014328 |
| GraphPad Prism 10 for macOS v.10.4.1 | GraphPad | SCR_002798 |
| Fiji (ImageJ2 v.2.9.0/1.53t) | https://fiji.sc/ | SCR_002285 |
| **Laboratory equipment** | | |
| Freedom EVO 200 robotic platform | Tecan Life Sciences | N/A |
| TROPHOS Plate Runner HD | TROPHOS | N/A |
| CM3050S cryostat | Leica | N/A |
| VT1000 S vibratome | Leica | N/A |
| SP5 confocal microscope | Leica | N/A |
| QuantStudio 3 Real-Time PCR System | Applied Biosystems | Cat#A28567 |
| **Other** | | |

## Cell lines

The NIH3T3 cell lines used in this study were previously described and generated from parental NIH3T3 cells that were obtained from the American Type Culture Collection (ATCC) without further authentication. NIH3T3-Smn$_{RNAi}$ cells express the tetracycline-dependent repressor (TetR) regulating the expression of an shRNA targeting endogenous mouse Smn driven by an H1$_{TO}$ promoter harboring two tandem copies of the Tet operator sequence (Ruggiu et al, 2012; Lotti et al, 2012). NIH3T3-SMN2/Smn$_{RNAi}$ cells were derived from NIH3T3-Smn$_{RNAi}$ to express low levels of RNAi-resistant human SMN from a stably integrated human *SMN2* gene (Li et al, 2013). NIH3T3-SmB$_{RNAi}$ cells express TetR and an shRNA targeting endogenous mouse SmB driven by the H1$_{TO}$ promoter and have normal SMN levels (Ruggiu et al, 2012). All NIH3T3 cell lines were cultured in Dulbecco's Modified Eagle's Medium (DMEM) with high glucose (Gibco) containing 10% fetal bovine serum (HyClone), 2 mM L-glutamine (Gibco), and 0.1 mg/ml gentamycin (GIBCO). Cells were regularly tested for mycoplasma using the Universal Mycoplasma Detection Kit (ATCC). RNAi was induced by the addition to the growth medium of Doxycycline HCl (Fisher Scientific) at the final concentration of 100 ng/ml.

## Cell proliferation assay

The cell proliferation assay in 96-well format was performed as previously described with minor modifications (Li et al, 2013). NIH3T3 cells were cultured with Dox for 5 days prior to seeding in 96-well optical plates (Greiner Bio-One) at 100 cells per well in 200 μl of Dox-containing media. Compounds were added 4 h post-plating, and experiments terminated after 5 additional days in culture by addition of a 4X solution containing 16% paraformaldehyde and 8 mg/ml of Hoechst in PBS to each well of the 96-well plate and further incubation for 15 min at room temperature. After washing with PBS, cell number determination was performed by whole-well imaging acquisition using a plate imaging system (TROPHOS Plate RUNNER HD), followed by nuclear counting with the TINA software (TROPHOS).

## Chemical compounds

For the chemical screen, we used a library of 1307 small molecules containing the complete Tocriscreen Mini (Tocris, #2890) collection and additional compounds from Tocris, highly enriched in FDA-approved drugs, known signaling inhibitors, and other bioactive molecules. SMN-C3 was custom synthesized by Combi-Block, Inc. according to the published compound structure (Naryshkin et al, 2014), dissolved at a concentration of 1.2 mg/ml in 100% DMSO, and stored in aliquots at −80 °C until use. MW150 was synthesized as previously described (Roy et al, 2015), dissolved at a concentration of 10 mg/ml in sterile 0.9% saline, and stored in aliquots at room temperature until use. All other compounds were obtained from Tocris Bioscience and dissolved in 100% DMSO unless otherwise specified.

## Cell-based chemical screen

A phenotypic screen for chemical compounds that promote cell proliferation of SMN-deficient mouse fibroblasts was carried out using Dox-treated NIH3T3-SMN2/Smn$_{RNAi}$ cells. A Tecan Freedom EVO 200 robotic platform equipped with a two-arm operation system and 96-channel pipetting stations was employed for liquid handling. Several parameters of the cell-based assay, including cell density, the time of Dox pre-treatment prior to plating and from compound addition to readout, inter-well and inter-plate variability, and edge effects were optimized prior to screening implementation. For the chemical screen, NIH3T3-SMN2/Smn$_{RNAi}$ cells were cultured with Dox for 5 days prior to seeding in 96-well optical plates (Greiner Bio-One) at 100 cells per well in 200 μl of Dox-containing media. About 1307 compounds from the Tocris library were tested at a single concentration (10 M) and one compound per well was added 4 h post-plating. In each 96-well plate, six wells were reserved for negative drug controls (0.5% DMSO) and six wells for positive drug controls (500 nM SMN-C3). Following 5 days of incubation, each well was scored using Hoechst staining and direct determination of cell number by imaging with the TROPHOS system. Any plate in which all of the positive control values did not exceed the range of negative control values were rejected. The primary screen was carried out in triplicate plates, and hits were defined as compounds causing a mean increase in cell number greater than the mean cell number of vehicle-treated cells plus 3 standard deviations. The screen yielded 15 compounds as initial hits, which underwent additional validation by (i) visual inspection of 96-well images to check for drug precipitates that may affect readout, (ii) repeat testing in multiple replicates, (iii) dose-response analysis, and iv) counter-screen assays. Primary hits were initially retested at 10 μM in six replicate wells using the same source of compounds and assay in NIH3T3-SMN2/Smn$_{RNAi}$ cells. Confirmed compounds were then repurchased from the vendor and analyzed in dose-response studies using concentrations spanning those used in the initial screening and six replicates per concentration in NIH3T3-SMN2/Smn$_{RNAi}$ cells. As counter-screen assays designed to reveal whether compounds promote cell proliferation in a non-specific manner,

compounds were also tested for their effects on proliferation using Dox-treated NIH3T3-SmB$_{RNAi}$ cells and NIH3T3-SMN2/Smn$_{RNAi}$ cells cultured with either normal (10%) or reduced (2%) serum in the absence of Dox. The half maximal effective concentration (EC50) was estimated by non-linear curve fitting using the dose-response data and the GraphPad Prism 10 software. Validated active compounds were defined as those with a reproducible activity and a potency of EC50 < 10 µM.

## Mouse lines

All mouse work was performed in accordance with the National Institutes of Health Guidelines on the Care and Use of Animals, complied with all ethical regulations and was approved by the Institutional Animal Care and Use Committee (IACUC) of Columbia University (AABR9600). Mice were housed in a 12 h/12 h light/dark cycle with access to food and water ad libitum. FVB.Cg-*Grm7*$^{Tg(SMN2)89Ahmb}$ *Smn1*$^{tm1Msd}$ Tg(SMN2*delta7)4299Ahmb/J (JAX Strain #005025) mice were interbred to obtain SMA mutant mice (Le et al, 2005). The *Smn*$^{2B/2B}$ mice on a pure FVB/N genetic background were previously described (Eshraghi et al, 2016). *Smn*$^{2B/2B}$ mice were crossed with heterozygous *Smn*$^{+/-}$ mice harboring the *Smn1*$^{tm1Msd}$ knockout allele (Schrank et al, 1997) on a pure FVB/N genetic background to generate *Smn*$^{2B/-}$ SMA mice and *Smn*$^{2B/+}$ control littermates as previously described (Carlini et al, 2022). Genotyping was performed from tail DNA as previously described for the SMNΔ7 (Gabanella et al, 2007) and *Smn*$^{2B}$ (Carlini et al, 2022) mouse lines. Equal proportions of mice of both sexes were used, and aggregated data were presented because gender-specific differences were not found.

## Drug treatments and behavioral analysis

SMA mice were treated daily via intraperitoneal (IP) injection with MW150 (5 mg/kg) starting at P0. SMN-C3 was administered IP daily at 3 mg/kg beginning at either P0 or P8 for early and delayed treatment, respectively. In the combinatorial treatment, SMA mice received 5 mg/kg MW150 starting at P0 and 3 mg/kg SMN-C3 from P8, with injections on alternating abdominal sides and at separate times. Mice from all experimental groups were monitored daily for survival and weight from birth for up to 30 days. Righting reflex was assessed by placing the mouse on its back and measuring the time it took to turn upright on its four paws (righting time). Cut-off test time was 60 s. For each testing session, the test was repeated three times, and the mean of the recorded times was calculated. The hindlimb suspension test was performed daily from P11 onward as previously described (Carlini et al, 2022). Litters were culled to 6 pups and randomly allocated to a specific drug treatment to avoid cross-contamination.

## MW150 biodistribution in mice

The exposure levels of MW150 in plasma and brain of SMA mice was measured following a single IP injection of four escalating drug concentrations (2.5, 5, 10, and 20 mg/kg) at P10. Brain and plasma samples were collected 3 h after injection from isoflurane anesthetized mice, quickly frozen, and shipped to Absorption Systems for determination of MW150 levels by a previously developed LC-MS/MS analytical method (Roy et al, 2015).

## RNA analysis

RNA from whole spinal cord tissue was extracted with TRIzol reagent (Invitrogen) and treated with DNase I (Ambion). RNA was reverse transcribed using the RevertAid first-strand cDNA kit (Thermo Scientific). Quantitative PCR (qPCR) was carried out using Power SYBR green PCR master mix (Applied Biosystems) in QuantStudio 3 (Applied Biosystems). The expression levels of mouse Smn mRNA, and total (SMN2$_{TOT}$) and full-length (SMN2$_{FL}$) human SMN mRNAs specifically transcribed from the *SMN2* gene were quantified as previously described (Ruggiu et al, 2012). A list of the primers used is shown in Table EV1.

## Protein analysis

For Western blot analysis, mice were sacrificed and spinal cord collection was performed in a dissection chamber under continuous oxygenation (95%O$_2$/5%CO$_2$) in the presence of cold (~12 °C) artificial cerebrospinal fluid (aCSF) containing 128.35 mM NaCl, 4 mM KCl, 0.58 mM NaH$_2$PO$_4$, 21 mM NaHCO$_3$, 30 mM D-Glucose, 1.5 mM CaCl$_2$, and 1 mM MgSO$_4$. Total protein extracts were generated by homogenization of spinal cords in SDS sample buffer (2% SDS, 10% glycerol, 5% ß-mercaptoethanol, 60 mM Tris-HCl pH 6.8, and bromophenol blue), followed by brief sonication and boiling. Proteins were quantified using the *RC DC*™ Protein Assay (Bio-Rad) and analyzed by SDS/PAGE on 12% polyacrylamide gels followed by Western blotting as previously described (Ruggiu et al, 2012). A list of the antibodies used is shown in Table EV2.

## Immunohistochemistry

For morphological studies by immunohistochemistry, mice were deeply anesthetized using Avertin, the depth of anesthesia was checked by the toe pinch reflex, and transcardial perfusion was then performed with a saline solution followed by 4% paraformaldehyde (PFA). The spinal cord and skeletal muscles were dissected and post-fixed by immersion in 4% PFA overnight at 4 °C. For immunohistochemistry, the spinal cords were briefly washed with PBS, and the L2 and L5 lumbar segments were identified by the ventral roots, dissected, and subsequently embedded in warm 5% agar. Transverse sections (75 µm) of the entire spinal segment were obtained with a VT1000 S vibratome (Leica). All the sections were then incubated overnight at room temperature with different combinations of primary antibodies diluted in PBS-T. The following day, six washing steps of 10 min each were done prior to incubation with secondary antibodies for 3 h in PBS. Another six washing steps were performed before sections were mounted in 30% glycerol/PBS. For NMJ analysis, skeletal muscles were cryoprotected through sequential immersion in 10% and 20% sucrose/0.1 M phosphate buffer (PB) for 1 h at 4 °C, followed by overnight immersion in 30% sucrose/0.1 M PB at 4 °C. The following day, muscles were frozen embedded in optimal cutting temperature (OCT) compound (Fisher), frozen on dry ice, and stored at −80 °C until processing. Longitudinal cryosections (30 µm) were collected onto Superfrost Plus glass slides (Fisher) using a CM3050S cryostat (Leica). Sections were washed once with PBS for 5 minutes to remove OCT, blocked for 1 h with 5% donkey serum in TBS containing 0.2% Triton-X at room temperature and incubated with primary antibodies in blocking buffer overnight at

4 °C. Following incubation, sections were washed three times for 10 min in TBS-T and then incubated with tetramethylrhodamine-conjugated α-bungarotoxin (Invitrogen #T1175, 1:500) and the appropriate secondary antibodies for 1 h at room temperature, followed by three washing steps. Slides were mounted using Fluoromount-G Mounting Medium (SouthernBiotech). A list of the antibodies used is shown in Table EV2.

### Confocal microscopy and image analysis

All images were collected with an SP5 confocal microscope (Leica) running the LAS AF software (v2.5.2.6939) and analyzed offline using the Leica LAS X software (v1.9.0.13747). For motor neuron number quantification, $1024 \times 1024$-pixel images were acquired from all the 75 μm sections of L2 and L5 spinal segments using a 20X objective at 3-μm steps in the z-axis and a 200 Hz acquisition rate. Only motor neurons (ChAT$^+$) with a clearly identifiable nucleus were counted to avoid double-counting from adjoining sections. For quantification of VGluT1$^+$ synapses, $1024 \times 1024$-pixel images were acquired from L2 spinal sections (75 μm) using a 40X objective at 0.3-μm steps in the z-axis and a 200 Hz acquisition rate. The total number of VGluT1$^+$ synapses per soma was determined by counting all the corresponding inputs over the entire surface of each ChAT$^+$ motor neuron cell body. At least ten motor neurons per mouse were quantified. For NMJ analysis, $1024 \times 1024$-pixel images were acquired from 30-μm muscle sections using a 20X objective at 2-μm steps in the z-axis and a 200 Hz acquisition rate. Maximum intensity projections of confocal stacks were analyzed, and at least 200 randomly selected NMJs per muscle were quantified. NMJs lacking any coverage of the α-bungarotoxin-labeled postsynaptic endplate by the presynaptic markers Synaptophysin and Neurofilament-M were scored as denervated, and those with less than 50% coverage were scored as partially innervated.

## Statistical analysis

The sample size for each experiment is detailed in the Figure legends and was determined based on previous publications. The investigators performing behavioral assays were blinded to genotype and treatment. All other experiments were performed unblinded. No data were excluded from the analysis. Results are expressed as mean and standard error of the mean (SEM) from at least three independent biological replicates unless otherwise indicated. The exact value and meaning of n for each dataset can be found in the Figure legends. The Shapiro–Wilk test was performed to check for the normality of data distribution. If data followed a parametric distribution, statistical analysis of two groups was performed by a two-tailed unpaired *t*-test when group variances were equal, and Welch's *t*-test when variances were unequal. One-way ANOVA followed by Tukey's multiple comparison test was used to compare three or more groups. Mann–Whitney test or Kruskal–Wallis followed by Dunn's multiple comparison test were applied when the assumptions for parametric tests were not met. Time-course comparisons of behavioral data (weight gain, righting time, and hindlimb suspension) were performed using a mixed-effects model ANOVA with the Geisser-Greenhouse correction. Comparison of survival curves was performed using the log-rank (Mantel–Cox) test. GraphPad Prism 10 for macOS Version 10.4.1 was used for all

**The paper explained**

**Problem**

Spinal muscular atrophy (SMA) is a severe neurodegenerative disease caused by low levels of the SMN protein, leading to the progressive loss of motor neurons and muscle weakness. Although current therapies aim to increase SMN levels and have improved outcomes, many patients continue to experience significant motor impairments. Since SMN deficiency disrupts multiple biological processes, there is growing interest in developing additional, SMN-independent treatments that target the downstream effects of the disease. Combining these with existing therapies may offer greater clinical benefit.

**Results**

We used a chemical screening approach to find compounds that could rescue the impaired cell growth caused by SMN deficiency. This led us to identify inhibitors of the p38 MAPK signaling pathway as potential therapeutic candidates. We showed that this pathway is activated by SMN loss in both cell cultures and in a mouse model of SMA. Treatment with MW150—a brain-penetrant, selective inhibitor of p38 MAPK—significantly improved motor function in SMA mice through SMN-independent neuroprotective effects. Notably, combining MW150 with a delayed SMN-enhancing therapy resulted in synergistic improvements in survival, motor function, and weight gain, far beyond either treatment alone. MW150's early protective action allowed the SMN therapy to later restore damaged neuromuscular and central synapses more effectively.

**Impact**

Our findings identify p38 MAPK as a promising therapeutic target in SMA and show that MW150 has strong potential as an SMN-independent neuroprotective agent. By preserving vulnerable motor neurons and enhancing synaptic repair, MW150 can amplify the benefits of existing SMN-inducing treatments. This supports the rationale for combining SMN restoration with targeted neuroprotection to improve the efficacy of SMA treatment.

statistical analyses, and *P* values are indicated as follows: *$P < 0.05$; **$P < 0.01$; ***$P < 0.001$; ****$P < 0.0001$.

## Data availability

This study includes no data deposited in external repositories.

The source data of this paper are collected in the following database record: biostudies:S-SCDT-10_1038-S44321-025-00303-6.

## Peer review information

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

## Acknowledgements

We are grateful to George Mentis for helpful discussions and critical reading of the manuscript. We thank Rashmi Kothary for providing the $Smn^{2B}$ mouse line. This work was supported by grants from CureSMA (L.P.) and NS071092 (L.P.), NS083831 (L.P.), NS098363 (L.P.), NS102451 (L.P.), NS114218 (L.P.), and NS116400 (L.P.) from NINDS.

## Author contributions

**Maria J Carlini**: Data curation; Formal analysis; Investigation; Visualization; Methodology; Writing—original draft; Writing—review and editing. **Jorge Espinoza-Derout**: Data curation; Investigation; Methodology; Writing—review and editing. **Meaghan Van Alstyne**: Data curation; Investigation; Methodology; Writing—review and editing. **Sarah Tisdale**: Data curation; Investigation; Methodology; Writing—review and editing. **Eileen Workman**: Data curation; Investigation; Methodology; Writing—review and editing. **Marina K Triplett**: Data curation; Investigation; Methodology; Writing—review and editing. **Ivan Tattoli**: Data curation; Investigation; Methodology; Writing—review and editing. **Shubhi Yadav**: Data curation; Investigation; Methodology; Writing—review and editing. **Christopher E Henderson**: Resources; Methodology; Writing—review and editing. **D Martin Watterson**: Resources; Methodology; Writing—review and editing. **Livio Pellizzoni**: Conceptualization; Data curation; Formal analysis; Supervision; Funding acquisition; Visualization; Writing—original draft; Project administration; Writing—review and editing.

Source data underlying figure panels in this paper may have individual authorship assigned. Where available, figure panel/source data authorship is listed in the following database record: biostudies:S-SCDT-10_1038-S44321-025-00303-6.

## Disclosure and competing interests statement

The authors declare no competing interests.

# Expanded View Figures

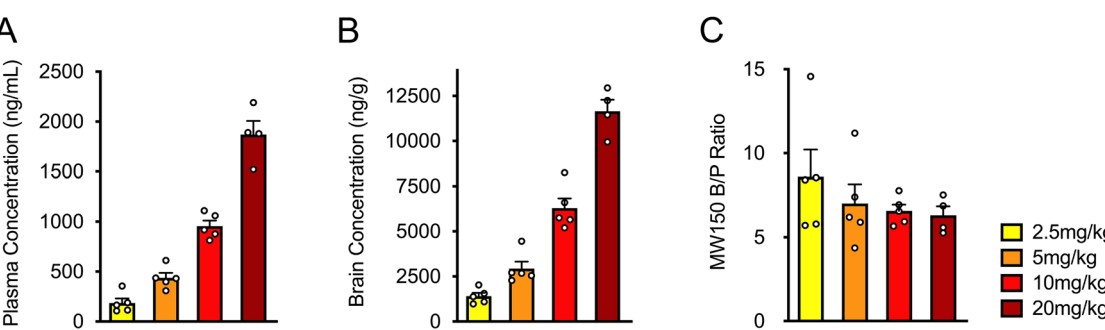

**Figure EV1. Analysis of MW150 biodistribution in plasma and brain of SMA mice.**

(A–C) MW150 concentration in plasma (A) and brain (B) and brain-to-plasma ratio (C) 3 h after a single IP injection of the indicated doses of MW150 in SMA mice at P10. Mean, SEM, and individual values from independent biological replicates (mice) for MW150 doses of 2.5 mg/kg ($n = 5$), 5 mg/kg ($n = 5$), 10 mg/kg ($n = 5$), and 20 mg/kg ($n = 4$) are shown. Source data are available online for this figure.

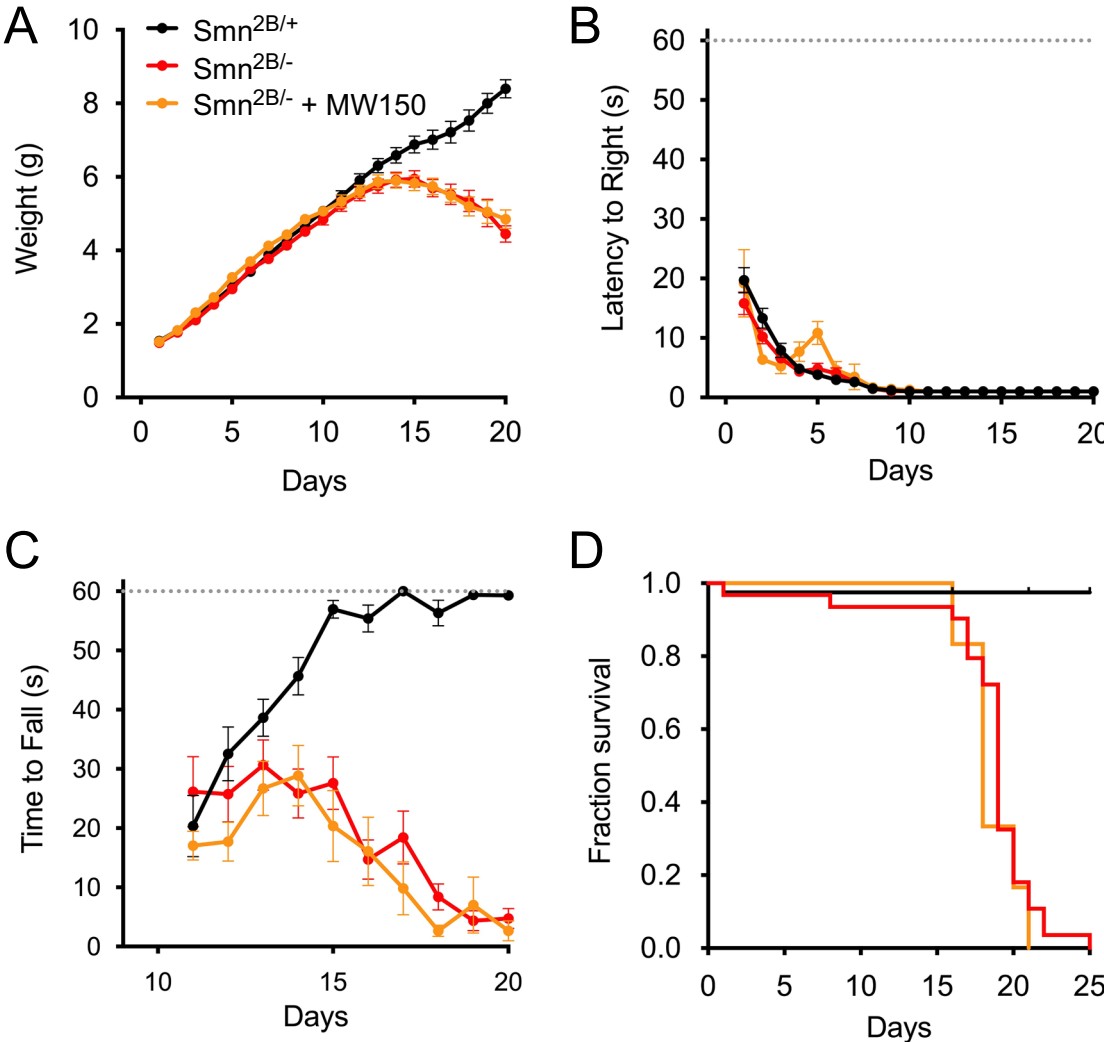

**Figure EV2. MW150 treatment does not improve the SMA phenotype in *Smn*[2B/−] mice.**

(A) Body weight of control *Smn*[2B/+] (n = 32) mice and *Smn*[2B/−] SMA mice either untreated (n = 31) or treated daily with MW150 (5 mg/kg) from P1 onward (n = 12). Data represent mean and SEM. Mixed-effects model ANOVA comparison of weight gain between *Smn*[2B/-] and *Smn*[2B/-] + MW150 mice: not significant. (B) Righting time from the same experimental groups shown in (A). Data represent mean and SEM. Mixed-effects model ANOVA comparison of righting time between *Smn*[2B/-] and *Smn*[2B/-] + MW150 mice: not significant. (C) Time to fall in the hindlimb suspension test from the same experimental groups shown in (A). Data represent mean and SEM. Mixed-effects model ANOVA comparison of hindlimb suspension test between *Smn*[2B/-] and *Smn*[2B/-] + MW150 mice: not significant. (D) Kaplan–Meier survival curves from the same experimental groups as in (A). Log-rank (Mantel–Cox) comparison of survival between *Smn*[2B/-] and *Smn*[2B/-] + MW150 mice: not significant. The data for untreated *Smn*[2B/+] and *Smn*[2B/-] mice are from a previously published study (Carlini et al, 2022). Source data are available online for this figure.

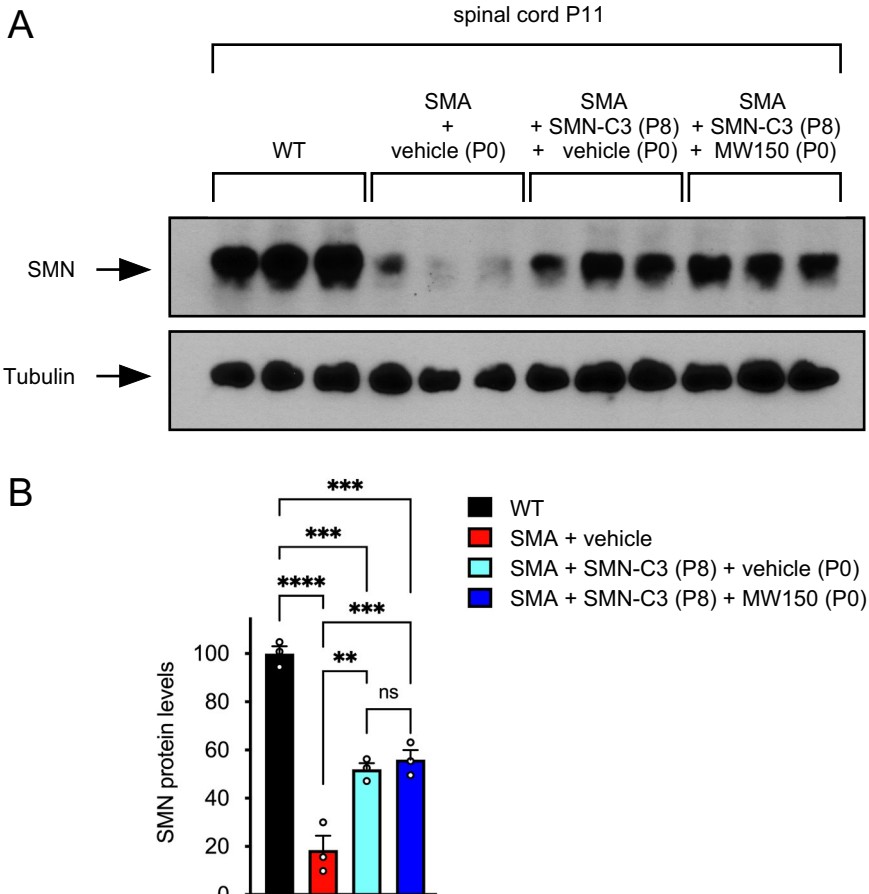

**Figure EV3. MW150 does not increase SMN expression beyond the levels induced by delayed treatment with SMN-C3 in the spinal cord of SMA mice.**

(A) Western blot analysis of SMN levels in P11 spinal cords from WT mice and SMA mice treated daily with vehicle or MW150 (5 mg/kg) starting at P0 and SMN-C3 (3 mg/kg) starting at P8 as indicated. Tubulin was used as loading control. (B) Quantification of SMN levels from the experiment in (A). Normalized mean, SEM, and individual values from three independent biological replicates are shown. One-way ANOVA and Tukey's post hoc test. $P < 0.0001$ (WT vs SMA+vehicle); $P = 0.0002$ (WT vs SMA + SMN-C3(P8) + vehicle(P0)); $P = 0.0003$ (WT vs SMA + SMN-C3(P8) + MW150(P0)); $P = 0.0019$ (SMA+vehicle vs SMA + SMN-C3(P8)+vehicle(P0)); $P = 0.0009$ (SMA + vehicle vs SMA + SMN-C3(P8) + MW150(P0)); ns not significant (SMA + SMN-C3(P8) + vehicle(P0) vs SMA + SMN-C3(P8) + MW150(P0)). Source data are available online for this figure.

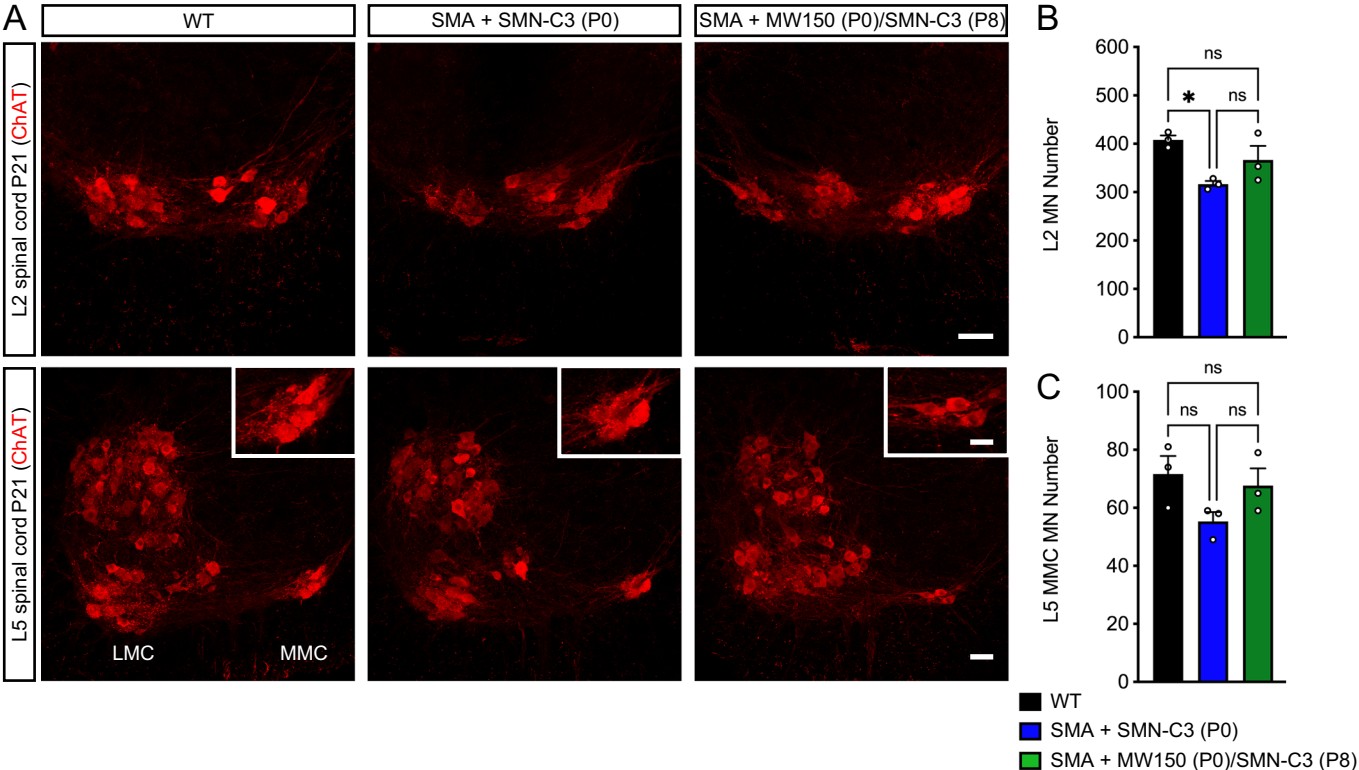

**Figure EV4.  Analysis of motor neuron survival in combinatorially treated adult SMA mice.**

(**A**) ChAT immunostaining of L2 and L5 spinal cords isolated at P21 from uninjected WT mice and SMA mice injected daily with SMN-C3 (3 mg/kg) starting at P0 or injected with SMN-C3 (3 mg/kg) at P8 in combination with MW150 (5 mg/kg) starting at P0. L5 LMC and MMC motor neuron pools are indicated, and magnified views of L5 MMC motor neurons are shown in the insets. Scale bars = 50 and 25 μm (insets). (**B**) Total number of L2 motor neurons in the same experimental groups as in (**A**). Normalized mean, SEM, and individual values from three mice per experimental group are shown. One-way ANOVA with Tukey's post hoc test. $P = 0.0259$ (WT vs SMA + SMN-C3(P0); ns, not significant. (**C**) Total number of L5 MMC motor neurons in the same experimental groups as in (**A**). Normalized mean, SEM, and individual values from three mice per experimental group are shown. One-way ANOVA with Tukey's post hoc test. ns, not significant. Source data are available online for this figure.

