## [Peer Review File · EMBO Molecular Medicine]

Identification of p38 MAPK inhibition as a neuroprotective strategy for combinatorial SMA therapy

Maria Carlini, Jorge Espinoza-Derout, Meaghan Van Alstyne, Sarah Tisdale, Eileen Workman, Marina Triplett, Ivan Tattoli, Shubhi Yadav, Christopher Henderson, D. Martin Watterson, and Livio Pellizzoni

Corresponding author(s): Livio Pellizzoni (lp2284@cumc.columbia.edu)

Review Timeline:

Submission Date:	18th Mar 25
Editorial Decision:	31st Mar 25
Revision Received:	17th Jul 25
Editorial Decision:	15th Aug 25
Revision Received:	18th Aug 25
Accepted:	20th Aug 25

Editor: Zeljko Durdevic

Transaction Report:

31st Mar 2025

Dear Prof. Pellizzoni,

Thank you for the submission of your manuscript to EMBO Molecular Medicine. We have now received feedback from the two reviewers who agreed to evaluate your manuscript. Both referees recognize interest of the study but also raise important concerns that should be addressed in a major revision. If you would like to discuss further the points raised by the referees, I am available to do so via email or video. Let me know if you are interested in this option.

We would welcome the submission of a revised version within three months for further consideration. Please let us know if you require longer to complete the revision.

I look forward to receiving your revised manuscript.

Yours sincerely,

Zeljko Durdevic

We require:

- 1) A .docx formatted version of the manuscript text (including legends for main figures, EV figures and tables). Please make sure that the changes are highlighted to be clearly visible.
- 2) Individual production quality figure files as .eps, .tif, .jpg (one file per figure). For guidance, download the 'Figure Guide PDF': (<https://www.embopress.org/page/journal/17574684/authorguide#figureformat>).
- 3) A .docx formatted letter INCLUDING the reviewers' reports and your detailed point-by-point responses to their comments. As part of the EMBO Press transparent editorial process, the point-by-point response is part of the Review Process File (RPF), which will be published alongside your paper.
- 4) A complete author checklist, which you can download from our author guidelines (<https://www.embopress.org/page/journal/17574684/authorguide#submissionofrevisions>). Please insert information in the checklist that is also reflected in the manuscript. The completed author checklist will also be part of the RPF.
- 5) Please note that all corresponding authors are required to supply an ORCID ID for their name upon submission of a revised manuscript.
- 6) It is mandatory to include a 'Data Availability' section after the Materials and Methods. Before submitting your revision, primary datasets produced in this study need to be deposited in an appropriate public database, and the accession numbers and

database listed under 'Data Availability'. Please remember to provide a reviewer password if the datasets are not yet public (see <https://www.embopress.org/page/journal/17574684/authorguide#dataavailability>).

12) Author contributions: You will be asked to provide CRediT (Contributor Role Taxonomy) terms in the submission system. These replace a narrative author contribution section in the manuscript.

13) A Conflict of Interest statement should be provided in the main text.

14) Every published paper now includes a 'Synopsis' to further enhance discoverability. Synopses are displayed on the journal webpage and are freely accessible to all readers. They include a short stand first (maximum of 300 characters, including space) as well as 2-5 one-sentences bullet points that summarizes the paper. Please write the bullet points to summarize the key NEW findings. They should be designed to be complementary to the abstract - i.e. not repeat the same text. We encourage inclusion of key acronyms and quantitative information (maximum of 30 words / bullet point). Please use the passive voice. Please attach

these in a separate file or send them by email, we will incorporate them accordingly.

15) Include a Reagents and Tools Table as part of the Methods section, which can be downloaded from our author guidelines (<https://www.embopress.org/page/journal/17574684/authorguide#structuredmethods>)

***** Reviewer's comments *****

Referee #1 (Remarks for Author):

In their work „Identification of p38 MAPK inhibition as a neuroprotective strategy for combinatorial SMA therapy" Carlini et al. use a cell line-based screening tool to identify small molecules which are able to modify phenotypes caused by the loss of the Survival Motor Neuron (SMN) protein. SMN loss leads to the motoneuron disease Spinal Muscular Atrophy (SMA) and using this strategy they identify p38 MAPK inhibitors as potential modifiers of the disease. Consistently, they show increased p38 activity in SMA mice spinal cords and an amelioration of the SMA-like phenotype using pharmacological p38 inhibition. Since current treatments are based on SMN-enhancing drugs with impressive yet limited effects, Carlini et al. established a delayed-intervention model of treated SMA using SMN-enhancing drugs. They then use this clinically relevant model for a combinatorial treatment approach using suboptimal, delayed-applied SMN-enhancing drugs together with a p38 inhibitor showing that the combination results in a better phenotype compared to the animals treated with SMN-enhancer alone. In subsequent analysis, they elucidate the underlying pathohistological mechanism showing that there may be synergistic action of p38-MAPK inhibition combined with SMN-enhancing on neuromuscular junction denervation while there seems to be no synergy on loss of central proprioceptive synapse loss on motoneuron cell bodies.

While the current work is to some degree redundant with previous work from the same group (Simon et al., 2019: p38 activation, amelioration of SMA-like phenotype in SMA mice using p38 inhibitors) other parts are new and highly innovative, specifically combinatorial approaches using SMN-enhancing drugs together with p38 inhibitors which is conceptually interesting and clinically highly relevant.

The work is very well-made despite some major points demanding clarification. I therefore suggest a major revision.

Major points

The introduction does not properly reflect the current state of knowledge on p38 in SMA. Please include literature on p38 activity in SMA and p38 inhibition as a treatment strategy into the introduction, specifically own work but also work from MacKenzie, and the Ebert/Ma lab.

Figure 1 and 2: Please clarify if data was used twice - Figure 1D (EO1428, SMN2SmnRNAi), and Figure 2 D (EO1428). If yes, avoid this.

Figure 4 and Figure 5: Please clarify if data was used twice - Figure 4G (black and red bars) compared to Figure 5 H (black and red bars) and Figure 4 H (black and red bars) compared to 5 I (black and red bars). If yes, avoid this.

Figure 5 and Figure 6: Please clarify if data was used twice - Figure 5A (turquoise group) compared to Figure 6B (black group). If yes, avoid this.

Please clarify the statistical entity ("the n") which was used to perform statistical tests (mice or motoneurons/NMJs) for Figures 4G,H; 5H,I; 7 B,D; 8B,D. Use bar graphs with dots with each dot representing a statistical independent "n" (=mouse) for clarification. Do not use statistical dependent entities such as motoneurons within a spinal cord.

Figure 6: Include an SMN blot comparing SMN levels in SMA+SMN-C3(P8) vehicle with SMA+SMN-C3(P8) MW150 to show that the synergistic/additive effect is independent of SMN protein levels.

Minor points

Clarify what "SMN-independent" means. Independent of SMN is not the same as "without an observable increase in SMN-protein". All observations detected in a model lacking SMN compared to a control can be regarded as SMN-dependent. In case of delayed intervention and from a clinical perspective it is more important which changes remain after restoring SMN-levels for a given treatment, treatment-timepoint and organ.

Figure1: Please, comment/discuss on Figure 1D: Why is there a decline of normalized cell number in SMN2/SmnRNAi cell number at higher concentrations while there is not such an effect in SmnRNAi cells?

Figure 1: Clarify why EO1428 lead you to p38. Is EO1428 an inhibitor of p38?

Figure 2: It would have been interesting to use MW150 together with SMN-C3 in vitro to see if there are synergistic/additive effects.

Figure 3: Delta7 mice are not able to properly right already at P1. How can they be pre-symptomatic? How is pre-symptomatic, onset etc. defined?

Figure 4A: Could you comment/discuss why SMA+MW150 has higher variability in motor function test compared to SMA+vehicle?

Figure4A: Which statistical test has been used to show significant difference between vehicle and MW150 treated animals?

The statement that Smn2B^{-/-} mice do not show motoneuron degeneration at all is too strong. Many groups show motoneuron degeneration in those mice and a comparison with other mouse models demonstrated motor neuron degeneration at least in L1 (Buettner et al. 2021). Additionally, consider the genetic background. The absence of an effect of MW150 treatment on 2B^{-/-} mice could indeed be due to less pronounced motoneuron death. However, technical aspects such as insensitive motor function tests (e.g. righting reflex is not a good test in these mice) could also apply. Moreover, the authors show an effect of MW150 treatment on central synapse number (Figure 7C,D) which the authors claim is also affected in 2B^{-/-} mice. Broadening the MW150 effects on more than one mouse model would increase significance of these findings. Vice versa, showing that this mechanism is relevant in a single mouse model only questions translatability/and significance of the findings.

Figure 4 or 5: It would be interesting to include a P-p38 blot in SMN-C3 delayed-restored mice to evaluate if increased P-p38 remains as a molecular target.

Figure 6: Which statistical tests have been used to demonstrate significance for effects reported in B, C, D.

Figure 7: Use a more hypothesis-driven post-test strategy instead of testing everything against everything. The most important comparisons are turquoise and yellow compared to green (to show 'synergy'). This is somehow lost. Also indicate non-significant comparisons into the figure in case they are important for interpretation of results.

It would have been interesting to count motoneurons in these groups as well.

Discussion:

Please check the terminology "synergy" since this -by pharmacological definitions- means an "over-additive" effect. Is this really the case?

Please discuss the experimental paradigm in figure 6 and what it clinically means that MW150 has been applied at P0 while SMN-C3 has been applied at P8. Please discuss the limitations with regard to translatability.

Method section:

Include a statement on randomization of mice before assigning them to a treatment group.

Include a statement on blinding of investigators in morphometric analysis according to genotype and treatment.

Referee #2 (Remarks for Author):

In this study, the authors perform a screen for compounds which can rescue a cell proliferation defect induced by depleted Smn levels. They have identified and validated 4 compounds 'hits' and suggest they act in an Smn independent manner. P38 MAPK inhibitors were prioritised for further study, and showed that a range of MAPK inhibitors show efficacy against the proliferation phenotype and none appear to increase Smn levels. The MAPK inhibitor MW150 was able to improve righting time, give slight increase in weight but had no effect on survival. It was also observed to increase MN number in L5 and L2 spinal cord. They then describe a model of early vs late SmnC3 administration, showing early administration improved righting, weight, survival but late administration has more limited benefits although both early and late increase Smn RNA and protein to similar levels. They give MW150 in combination with late Smn upregulation and show synergy in righting, weight and survival. They also show MW150 can act in synergy with delayed Smn upregulation to further reduce denervation but not de-afferentation. However, extending the time course to P21 revealed afferent input numbers in late SmnC3 + MW150 were comparable to early SmnC3 and control mice.

Overall this is an interesting and potentially impactful study. The screening methodology is innovative and the targets identified novel for this disease. The persistent pathology following Smn upregulation is a real problem in the SMA field and ability to ameliorate this are extremely valuable. Targeting P38MAPK is novel and the ability to ameliorate motor neuron loss and act in synergy with Smn upregulators is potentially extremely valuable. The experiments are mostly performed to a very high standard models employed have a high translational value.

I have the following specific comments:

The initial drug screen was designed to look for compounds which can ameliorate a proliferation defect in Smn KD NIH3T3 cells. I can appreciate the logic in using this read out if the screen were designed to look for Smn up-regulators, but, since defects in proliferation are not observed in people with SMA or animal models, it seems a high risk strategy when looking for Smn independent modifiers. Can the authors comment on the logic here?

It seems surprising in such a large screen than no compounds were identified which did increase Smn levels. The authors comment that 3 of the 4 'hits' did not promote proliferation in cell lines lacking SMN2, but none of the 4 compounds were able to increase levels of Smn. Again I find this curious. Can the authors comment on the possible MOA, considering it is Smn independent, but not evident when SMN2 is absent?

In an extension of this, if the authors wish to conclude that the effects of the 4 hits are Smn independent, as they do form the data in figure 2, then Smn protein levels need to be properly quantified.

It seems remarkable that a compound identified as helping remedy proliferation is also neuroprotective. Can the authors comment on the potential shared mechanism between these processes that their drug is exploiting.

Page 10/11, the authors report that MW150 did not improve motor function in the Smn2B⁻ mouse model of SMA. They suggest that this is evidence of MF150 acting directly on cell death pathways in motor neurons as there is no motor neuron death in this mouse model. This is not an accepted view point in the field. Motor neuron loss in this model has been reported in a number of studies (e.g. PMID: 39973467; PMID: 36089582; PMID: 33382987; PMID: 32828271; PMID: 31365319; PMID: 31273192; PMID: 22071333) and includes one study implicating P38MAPK in the process (PMID: 37953591). Since motor neuron number and motor unit pathology was not quantified here, and the motor tests used here are very coarse measures of motor ability, the authors cannot make this conclusion.

The data in Figure 7 and Figure 8 regarding a the number of proprioceptive inputs considers a replicate to be a single motor neuron. Since I assume many motor neurons come from 1 animal, the statistical outcomes are potentially confounded by a pseudoreplication bias. The data analysis to be adjusted to correct for this.

In Figure 8, the authors report that the proprioceptive synapse defect, which was not fixed by P11 was remedied by P21. Is it possible that between P11 and P21 there has been a loss of the motor neurons which have the decreased number of proprioceptive inputs, meaning at P21 there is a preferential bias towards healthier motor neurons. Motor neuron counts should be done to confirm.

Can the authors comment on other literature which suggest that activation of P38 MAPK increases Smn levels? PMID: 19648294

Response to Reviewers – EMM-2025-21654, Carlini et al.**Point by point response****Reviewer #1:**

In their work „Identification of p38 MAPK inhibition as a neuroprotective strategy for combinatorial SMA therapy" Carlini et al. use a cell line-based screening tool to identify small molecules which are able to modify phenotypes caused by the loss of the Survival Motor Neuron (SMN) protein. SMN loss leads to the motoneuron disease Spinal Muscular Atrophy (SMA) and using this strategy they identify p38 MAPK inhibitors as potential modifiers of the disease. Consistently, they show increased p38 activity in SMA mice spinal cords and an amelioration of the SMA-like phenotype using pharmacological p38 inhibition. Since current treatments are based on SMN-enhancing drugs with impressive yet limited effects, Carlini et al. established a delayed-intervention model of treated SMA using SMN-enhancing drugs. They then use this clinically relevant model for a combinatorial treatment approach using suboptimal, delayed-applied SMN-enhancing drugs together with a p38 inhibitor showing that the combination results in a better phenotype compared to the animals treated with SMN-enhancer alone. In subsequent analysis, they elucidate the underlying pathohistological mechanism showing that there may be synergistic action of p38-MAPK inhibition combined with SMN-enhancing on neuromuscular junction denervation while there seems to be no synergy on loss of central proprioceptive synapse loss on motoneuron cell bodies.

While the current work is to some degree redundant with previous work from the same group (Simon et al., 2019: p38 activation, amelioration of SMA-like phenotype in SMA mice using p38 inhibitors) other parts are new and highly innovative, specifically combinatorial approaches using SMN-enhancing drugs together with p38 inhibitors which is conceptually interesting and clinically highly relevant.

The work is very well-made despite some major points demanding clarification. I therefore suggest a major revision.

We thank the reviewer for their overall positive evaluation of our study and have addressed the points raised as described below.

Major points

The introduction does not properly reflect the current state of knowledge on p38 in SMA. Please include literature on p38 activity in SMA and p38 inhibition as a treatment strategy into the introduction, specifically own work but also work from MacKenzie, and the Ebert/Ma lab.

We included comments on the work on p38 MAPK from the MacKenzie lab (which was also suggested by Reviewer #2) in the Introduction (page 5) and in the Discussion (page 20) where they seemed most appropriate.

To our knowledge, there are no studies from the Ebert and Ma labs, either separately or together, reporting any findings related to p38 MAPK in SMA. We think the Reviewer is possibly referring to a recent PNAS paper from their groups (PMID: 37976261). However, this study focuses on the p35 subunit of Cdk5 not p38 MAPK.

Figure 1 and 2: Please clarify if data was used twice - Figure 1D (EO1428, SMN2SmnRNAi), and Figure 2 D (EO1428). If yes, avoid this.

The same data for EO1428 from Figure 1D was included in Figure 2D for ease of comparison with the other p38MAPK inhibitors. As requested, a different dataset is now presented in Figure 2H of the revised manuscript.

Figure 4 and Figure 5: Please clarify if data was used twice - Figure 4G (black and red bars) compared to Figure 5 H (black and red bars) and Figure 4 H (black and red bars) compared to 5 I (black and red bars). If yes, avoid this.

Yes, we used the same data from untreated WT mice and vehicle-treated SMA mice in Figures 4G,H and 5H,I to facilitate direct comparison of this type of analysis among different figures. We have replaced the data for WT and SMA+vehicle groups in Figure 4G,H of the revised manuscript. The conclusions are the same of the original manuscript.

Figure 5 and Figure 6: Please clarify if data was used twice - Figure 5A (turquoise group) compared to Figure 6B (black group). If yes, avoid this.

Yes, the behavioral data for the SMN-C3 (P8) group is the same in both figures. We see no reason why the data could not be reused for comparison with a different experimental group. Repeating the experiment with additional 15 mice is also ethically unjustified as the use of vertebrate mice should be minimized. We clarify in the legend of Figure 6 that the SMNC3 (P8) dataset is the same of Figure 5A-C.

Please clarify the statistical entity ("the n") which was used to perform statistical tests (mice or motoneurons/NMJs) for Figures 4G,H; 5H,I; 7 B,D; 8B,D. Use bar graphs with dots with each dot representing a statistical independent "n" (=mouse) for clarification. Do not use statistical dependent entities such as motoneurons within a spinal cord.

As requested, we have replaced all the box-and-whiskers plots with bar graphs including dots for each independent experimental value (per mouse average). The legends indicate the sample size and the statistical entity ("the n") used to perform statistical tests, which is "mice" for all the morphological parameters analyzed (motor neuron number, NMJ innervation, and synaptic coverage). Please note that, as indicated in the legends of the original manuscript, VGlut1 synaptic coverage was the only parameter for which we used the number of synapses on individual motor neurons rather than mice as the "n" (Figures 7D and 8D) because it better reflects physiological (and pathological) differences in the number of proprioceptive synapses on individual motor neurons. As requested, however, the VGlut1 data are now presented as per mouse average in the revised manuscript. The conclusions are the same of the original manuscript.

Figure 6: Include an SMN blot comparing SMN levels in SMA+SMN-C3(P8) vehicle with SMA+SMN-C3(P8) MW150 to show that the synergistic/additive effect is independent of SMN protein levels.

We have performed the requested experiment, and the new data are shown in Figure EV3. As expected from earlier data shown in Figure 4D-E, combination treatment of MW150 with SMN-C3 (P8) does not further increase SMN levels relative to treatment with SMN-C3 (P8) alone in the spinal cord of SMA mice, indicating that the synergistic effects are independent of SMN levels.

Minor points

Clarify what "SMN-independent" means. Independent of SMN is not the same as "without an observable increase in SMN-protein". All observations detected in a model lacking SMN

compared to a control can be regarded as SMN-dependent. In case of delayed intervention and from a clinical perspective it is more important which changes remain after restoring SMN-levels for a given treatment, treatment-timepoint and organ.

We edited the text of the revised manuscript to clarify the issue and replaced “SMN-independent” with “independent of SMN induction”.

Figure1: Please, comment/discuss on Figure 1D: Why is there a decline of normalized cell number in SMN2/SmnRNAi cell number at higher concentrations while there is not such an effect in SmnRNAi cells?

Depending on the compound, there is also a detectable albeit less pronounced decline in Smn_{RNAi} cells. As we previously reported in similar experiments using valproic acid (PMCID: PMC3744461), differences in the decline in normalized cell number at higher (toxic) compound concentrations likely reflects the effects of toxicity combined with the greater proliferation rate of SMN2/Smn_{RNAi} cells relative to Smn_{RNAi} cells. We included a comment in the revised manuscript (page 7).

Figure 1: Clarify why EO1428 lead you to p38. Is EO1428 an inhibitor of p38?

Yes, EO1428 is a small molecule inhibitor of p38 MAPK. We clarified this point and added a reference in the revised manuscript (page 8).

Figure 2: It would have been interesting to use MW150 together with SMN-C3 in vitro to see if there are synergistic/additive effects.

We appreciate the suggestion but did not perform the experiments because, irrespective of the outcome, they would not provide mechanistic insights without performing additional functional studies in the cellular model.

Figure 3: Delta7 mice are not able to properly right already at P1. How can they be pre-symptomatic? How is pre-symptomatic, onset etc. defined?

We agree with the Reviewer and modified our wording from pre-, early-, and late-symptomatic to early-, mid-, and late-symptomatic, respectively (page 9).

Figure 4A: Could you comment/discuss why SMA+MW150 has higher variability in motor function test compared to SMA+vehicle?

Vehicle treated SMA mice display a quite uniform inability to right themselves across their short lifespan. Treatment with MW150 partially improves motor function (righting time) by targeting some (i.e. motor neuron death) but not all of the underlying deficits (i.e. synaptic loss, etc) associated with this phenotype and the overall SMA pathology. Accordingly, the disease continues to progress in MW150 treated SMA mice leading to the erosion of the gain in motor function, which is most prominent one or two days prior to death when mice are overtly sick and often paralyzed. Accordingly, the higher variability in the average righting time shown in Figure 4A reflects differences in the time when individual MW150-treated SMA mice reach disease end stage in the second postnatal week. We added a comment in the revised manuscript (page 10).

Figure4A: Which statistical test has been used to show significant difference between vehicle and MW150 treated animals?

We apologize for the oversight and have included the results and statistical tests used to compare behavioral data in the methods section and in the figure legends of these (Figure 4A-C) as well as other similar experiments (Figures 5A-C, 6B-D, EV2A-D).

The statement that *Smn*^{2B/-} mice do not show motoneuron degeneration at all is too strong. Many groups show motoneuron degeneration in those mice and a comparison with other mouse models demonstrated motor neuron degeneration at least in L1 (Buettner et al. 2021). Additionally, consider the genetic background. The absence of an effect of MW150 treatment on 2B/- mice could indeed be due to less pronounced motoneuron death. However, technical aspects such as insensitive motor function tests (e.g. righting reflex is not a good test in these mice) could also apply. Moreover, the authors show an effect of MW150 treatment on central synapse number (Figure 7C,D) which the authors claim is also affected in 2B/- mice. Broadening the MW150 effects on more than one mouse model would increase significance of these findings. Vice versa, showing that this mechanism is relevant in a single mouse model only questions translatability/and significance of the findings.

We appreciate the Reviewer's comments, which are also shared by Reviewer #2, and have revised the text in a manner that we believe adequately describes our findings, taking into account the Reviewers' critiques and striking a balance between different views in the field (page 11). We agree with both Reviewers that the available tests applicable to mice during early postnatal development are coarse measures of motor function and have now disclosed it as a potential limitation of our analysis.

We feel that the most contentious point (i.e. differences in the extent of motor neuron death in *Smn*^{2B/-} SMA mice) does not merit further debate within the current manuscript because we have extensively addressed the potential reasons for the discrepancy between our previously published findings (PMID: 35913953) and those from others (PMID: 28172892; PMID: 31273192; PMID: 33382987; PMID: 36089582). These include i) the sensitivity of methods employed for motor neuron counting (section sampling versus actual counts of the total number of motor neurons per spinal segment); and ii) the analysis of specific spinal segments versus pooled analysis from sections spanning multiple segments that differ in both the number of motor neurons they contain and their susceptibility to disease.

The genetic background is unlikely a confounding factor contributing to the different results regarding motor neuron loss reported in our study (PMID: 35913953) relative to others that employed *Smn*^{2B/-} mice in the same FVB/N genetic background (PMID: 28172892).

Buettner et al (PMID: 34825141) employed *Smn*^{2B/-} mice in the C57BL/6 background and a method to count motor neurons that is analogous to ours. These authors did not find any loss of SMA motor neurons in the T9 and L5 (either for LMC or MMC motor neurons) spinal segments and a modest loss of L1 motor neurons at a very late time point (P26), which is past the median survival (P24) that is only reached by about 20% of *Smn*^{2B/-} SMA mice. This modest and very late-onset loss of motor neurons – restricted to a single spinal segment documented by Buettner et al (PMID: 34825141) – is in striking contrast with the wider loss of thoracic and lumbar motor neurons reported to occur at much earlier times (P15-P16) in other studies using *Smn*^{2B/-} mice in the C57BL/6 background (PMID: 28172892; PMID: 31273192; PMID: 33382987; PMID: 36089582). Lastly, there is also a lack of consistency among studies reporting loss of motor neurons in *Smn*^{2B/-} mice, with lumbar motor neurons being robustly lost in some studies (PMID: 28172892; PMID: 33382987) but entirely spared in another (PMID: 36089582) despite analyses being performed in the same mouse model and age.

Overall, our previous findings (PMID: 35913953) and those from Buettner et al (PMID: 34825141) are well-aligned to support the conclusion that motor neuron death is not a

prominent contributor to SMA pathology in $Smn^{2B/-}$ mice. At a minimum, we argue that there is not an accepted view in the field regarding the death of motor neurons in $Smn^{2B/-}$ mice.

Regarding the issue of translatability of our findings in $SMN\Delta 7$ mice to other SMA models, the Reviewer states correctly that proprioceptive synapses are affected in $Smn^{2B/-}$ SMA mice, as we and others have previously demonstrated. However, contrary to the Reviewer's statement, we clearly show that MW150 treatment alone does not improve the loss of these central synapses in SMA mice (Figure 7C,D). The reported increase in proprioceptive synaptic coverage is driven by SMN induction (not MW150) and only occurs upon combination treatment with MW150 and delayed SMN-C3 at later times (P21) (Figure 8C,D). Importantly, unlike $SMN\Delta 7$ mice, the $Smn^{2B/-}$ mouse model does not harbor the human *SMN2* gene and is not suitable for testing *SMN2* splicing modifying therapies. Therefore, there is no rationale for testing combinatorial therapy with MW150 and SMN-C3 in $Smn^{2B/-}$ SMA mice. This is an intrinsic experimental limitation of the $Smn^{2B/-}$ model for such combinatorial studies, which has no bearing on the translatability or general significance of our findings.

Figure 4 or 5: It would be interesting to include a P-p38 blot in SMN-C3 delayed-restored mice to evaluate if increased P-p38 remains as a molecular target.

The proposed analysis would not be informative because there are no significant differences in the levels of phosphorylated p38 MAPK in the spinal cord of WT and SMA mice at P11 (Figure 3E-F).

Figure 6: Which statistical tests have been used to demonstrate significance for effects reported in B, C, D.

See above for our response to the same point raised for Figure 4A.

Figure 7: Use a more hypothesis-driven post-test strategy instead of testing everything against everything. The most important comparisons are turquoise and yellow compared to green (to show 'synergy'). This is somehow lost. Also indicate non-significant comparisons into the figure in case they are important for interpretation of results.

We have revised the graphs according to the Reviewer's suggestion.

It would have been interesting to count motoneurons in these groups as well.

Motor neuron counts at P11 are shown for all experimental groups except for combined treatment with MW150 (P0) and SMNC3 (P8) in Figures 4F-H and 5G-I. In the revised manuscript, and also as part of our response to Reviewer #2 on a related point, we have added new data for motor neuron counts at P21 including in SMA mice following combination treatment (Figure EV4).

Discussion:

Please check the terminology "synergy" since this -by pharmacological definitions- means an "over-additive" effect. Is this really the case?

We believe the behavioral data in Figure 6 provide compelling evidence for the synergistic effects of the combined use of MW150 and SMNC3 (P8) relative to each individual treatment. Both the magnitude of the behavioral improvements and the persistence of the effects of combinatorial treatment well past the median survival of mice when treated with each drug alone support this conclusion. Moreover, while neither treatment alone provides any benefit on weight gain and survival in SMA mice, their combined use shows strong enhancement.

Please discuss the experimental paradigm in figure 6 and what it clinically means that MW150 has been applied at P0 while SMN-C3 has been applied at P8. Please discuss the limitations with regard to translatability.

We addressed this comment in the Discussion (page 21).

Method section:

Include a statement on randomization of mice before assigning them to a treatment group.

We included the information in the Methods section (page 27).

Include a statement on blinding of investigators in morphometric analysis according to genotype and treatment.

We included the information in the Methods section (page 30).

Reviewer #2:

In this study, the authors perform a screen for compounds which can rescue a cell proliferation defect induced by depleted Smn levels. They have identified and validated 4 compounds 'hits' and suggest they act in an Smn independent manner. P38 MAPK inhibitors were prioritised for further study, and showed that a range of MAPK inhibitors show efficacy against the proliferation phenotype and none appear to increase Smn levels. The MAPK inhibitor MW150 was able to improve righting time, give slight increase in weight but had no affect on survival. It was also observed to increase MN number in L5 and L2 spinal cord. They then describe a model of early vs late SmnC3 administration, showing early administration improved righting, weight, survival but late administration has more limited benefits although both early and late increase Smn RNA and protein to similar levels. They give MW150 in combination with late Smn upregulation and show synergy in righting, weight and survival. They also show MW150 can act in synergy with delayed Smn upregulation to further reduce denervation but not de-afferentation. However, extending the time course to P21 revealed afferent input numbers in late SmnC3 + MW150 were comparable to early SmnC3 and control mice.

Overall this is an interesting and potentially impactful study. The screening methodology is innovative and the targets identified novel for this disease. The persistent pathology following Smn upregulation is a real problem in the SMA field and ability to ameliorate this are extremely valuable. Targeting P38MAPK is novel and the ability to ameliorate motor neuron loss and act in synergy with Smn upregulators is potentially extremely valuable. The experiments are mostly performed to a very high standard models employed have a high translational value.

We thank the reviewer for their appreciation of the novelty, interest, and impact of our study.

I have the following specific comments:

The initial drug screen was designed to look for compounds which can ameliorate a proliferation defect in Smn KD NIH3T3 cells. I can appreciate the logic in using this read out if the screen were designed to look for Smn up-regulators, but, since defects in proliferation are not observed in people with SMA or animal models, it seems a high risk strategy when looking for Smn independent modifiers. Can the authors comment on the logic here?

The logic was that an unbiased phenotypic screen in our model system could capture both SMN inducers and compounds acting independently of SMN induction. For example, hit compounds could be molecules stimulating its ubiquitous activity in snRNP assembly that is disrupted in

both fibroblasts and animal models of the disease or modify proximal downstream targets of its deficiency that are not restricted to fibroblasts. Of course, the approach might also lead to the identification of compounds acting further downstream the cascade of events induced by SMN deficiency that specifically benefit proliferation deficits in fibroblasts but have lesser disease relevance. While addressing mechanism of action and disease relevance is the objective of follow up studies, we felt that the risk associated with our screening approach was well balanced by the high potential reward for discovery of unanticipated modifiers. As it turns out, through this screening approach we identified p38 MAPK activation as a shared event downstream of SMN deficiency in both in vitro and in vivo models.

It seems surprising in such a large screen than no compounds were identified which did increase *Smn* levels. The authors comment that 3 of the 4 'hits' did not promote proliferation in cell lines lacking SMN2, but none of the 4 compounds were able to increase levels of *Smn*. Again I find this curious. Can the authors comment on the possible MOA, considering it is *Smn* independent, but not evident when SMN2 is absent?

Having found no compounds acting directly on SMN expression might seem surprising, but we wish to note that the size of the compound library we tested was not particularly large for high-throughput screening's standards. Except for p38 MAPK inhibition, which is followed up in the study, we have no specific elements for suggesting possible mechanisms of action for the other compounds on cell proliferation and prefer not to speculate on this aspect. The lower activity of the other 3 "hits" relative to that of EO1428 may explain their muted effects in *Smn*_{RNAi} cells whose proliferation is more severely impaired than SMN2/*Smn*_{RNAi} cells. We added a comment in the revised manuscript (page 8).

In an extension of this, if the authors wish to conclude that the effects of the 4 hits are *Smn* independent, as they do from the data in figure 2, then *Smn* protein levels need to be properly quantified.

We thank the Reviewer for their comment and have added new data to the revised manuscript that thoroughly addresses this point. We performed RT-qPCR experiments and measured the effects of the 4 "hits" on the expression of endogenous mouse *Smn* mRNA to test for potential interference with the RNAi system as well as total and exon 7 containing full-length human SMN mRNAs to test for transcriptional and/or splicing induction (new Figure 2A-C). We also included quantification of SMN protein levels from western blot experiments (new Figure 2E). Overall, we found no significant changes in SMN levels at the mRNA or protein levels in compound-treated, *Smn*-deficient NIH3T3-SMN2/*Smn*_{RNAi} cells relative to DMSO-treated controls.

It seems remarkable that a compound identified as helping remedy proliferation is also neuroprotective. Can the authors comment on the potential shared mechanism between these processes that their drug is exploiting.

The shared mechanism(s) is inhibition of p38 MAPK kinase, which we show is activated both in vitro (Figure 2BF-G) and in vivo (Figure 3). The downstream events are then likely different in proliferating fibroblasts and post-mitotic motor neurons. While we don't find strong value in dissecting these downstream mechanisms affecting cell proliferation in the NIH3T3 model, which may not be disease relevant as also pointed out by this Reviewer, we have linked p38 MAPK activation to phosphorylation of p53 in SMA motor neurons (PMCID: PMC6956708). The latter point is discussed in page 19.

Page 10/11, the authors report that MW150 did not improve motor function in the *Smn*2B/- mouse model of SMA. They suggest that this is evidence of MF150 acting directly on cell death

pathways in motor neurons as there is no motor neuron death in this mouse model. This is not an accepted view point in the field. Motor neuron loss in this model has been reported in a number of studies (e.g. PMID: 39973467; PMID: 36089582; PMID: 33382987; PMID: 32828271; PMID: 31365319; PMID: 31273192; PMID: 22071333) and includes one study implicating P38MAPK in the process (PMID: 37953591). Since motor neuron number and motor unit pathology was not quantified here, and the motor tests used here are very course measures of motor ability, the authors cannot make this conclusion.

Please see our response to the similar point raised by Reviewer #1.

We acknowledge that there are different viewpoints in the field. However, we stand by the results published in our previous study (PMID: 35913953) and respectfully argue against the relevance of counting motor neurons in *Smn*^{2B/-} mice treated with MW150 as we found no evidence for significant neuronal loss in this SMA model relative to normal controls.

The data in Figure 7 and Figure 8 regarding a the number of proprioceptive inputs considers a replicate to be a single motor neuron. Since I assume many motor neurons come from 1 animal, the statistical outcomes are potentially confounded by a psuedoreplication bias. The data analysis to be adjusted to correct for this.

As requested, we have replaced the graphs relative to the quantification of the number of VGluT1⁺ synapses on motor neurons with the corresponding averages from each mouse in Figures 7D and 8D. The conclusions are the same of the original manuscript. See also our response to Reviewer #1 related to this point.

In Figure 8, the authors report that the proprioceptive synapse defect, which was not fixed by P11 was remedied by P21. Is it possible that between P11 and P21 there has been a loss of the motor neurons which have the decreased number of prioprioceptive inputs, meaning at P21 there is a preferential bias towards healthier motor neurons. Motor neuron counts should be done to confirm.

Previous studies have shown that motor neuron death is independent from the loss of proprioceptive synapses and that the latter is a direct consequence of the effect of SMN deficiency in proprioceptive neurons. Accordingly, restoration of VGluT1⁺ afferents on SMA motor neurons can occur without any improvement in motor neuron survival (PMID: 28504671; PMID: 32516136). Conversely, selective rescue of motor neuron survival does not improve deafferentation SMA mice (PMID: 28504671; PMID: 30012555; PMID: 29281826). Together, this evidence argues against the possibility raised by the Reviewer.

Nevertheless, to address directly this point and a similar one raised by Reviewer #1, we performed the requested analysis of motor neuron counts for the experimental groups in Figure 8. The new data is presented in Figure EV4. The results show that the number of motor neurons in SMA mice treated with MW50 and SMN-C3 (P8) is not significantly lower than in WT mice, arguing against the loss of vulnerable motor neurons at later times as a contributing factor for the observed increase in VGluT1 synapses.

Can the authors comment on other literature which suggest that activation of P38 MAPK increases *Smn* levels? PMID: 19648294

We addressed the work on p38 MAPK from the MacKenzie lab in the Introduction (page 5) and in the Discussion (page 20) as also suggested by Reviewer #1.

15th Aug 2025

Dear Prof. Pellizzoni,

Thank you for the submission of your revised manuscript to EMBO Molecular Medicine and please accept my apologies for the delay in getting back to you due to the holiday season. I am pleased to inform you that we will be able to accept your manuscript pending the following final amendments:

- 1) Please implement the referee suggestion. The central synapse counts should be moved to a new EV figure.
- 2) Please address all comments suggested by our data editors listed below:
 - o Figure legends:
 1. Please note that the legend for figure 3 is not provided in the sequential manner. This needs to be rectified.
 2. Please note that the exact p values are not provided in the legends of figures 2G; 3D, 4E, G, H; 5D, F, H, I; 7B, D; EV3 B, EV4 B, C.
 3. Please note that information related to n is missing in the legends of figures EV1 A-C.
 - Limit keywords to max. 5.
 - Rename "Materials and Methods" to "Methods"
 - Author contributions: Please remove it from the manuscript and specify author contributions in our submission system. CRediT has replaced the traditional author contributions section because it offers a systematic machine-readable author contributions format that allows for more effective research assessment. You are encouraged to use the free text boxes beneath each contributing author's name to add specific details on the author's contribution. More information is available in our guide to authors:
<https://www.embopress.org/page/journal/17574684/authorguide#authorshipguidelines>
 - Indicate in legends exact n and exact p values, not a range, along with the statistical test used. To keep the figures "clear" some authors found providing an Appendix table Sx with all exact p-values preferable. You are welcome to do this if you want to.
 - In data availability statement replace the current text with "This study includes no data deposited in external repositories".
- 3) Appendix: Please remove Appendix, upload the tables as separate files and rename them to Table EV1 and EV2. Also, update their callouts in the main text.
- 4) The Paper Explained: Please add it to the main manuscript file.
- 5) Synopsis: Please check your synopsis text and image before submission with your revised manuscript. Please be aware that in the proof stage minor corrections only are allowed (e.g., typos).
- 6) As part of the EMBO Publications transparent editorial process initiative (see our Editorial at <http://embomolmed.embopress.org/content/2/9/329>), EMBO Molecular Medicine will publish online a Review Process File (RPF) to accompany accepted manuscripts. This file will be published in conjunction with your paper and will include the anonymous referee reports, your point-by-point response and all pertinent correspondence relating to the manuscript. Let us know whether you agree with the publication of the RPF and as here, if you want to remove or not any figures from it prior to publication. Please note that the Authors checklist will be published at the end of the RPF.
- 7) Please provide a point-by-point letter INCLUDING my comments as well as the reviewer's reports and your detailed responses (as Word file).

I look forward to reading a new revised version of your manuscript as soon as possible.

Yours sincerely,

Zeljko Durdevic

Zeljko Durdevic
Senior Editor
EMBO Molecular Medicine

*** Instructions to submit your revised manuscript ***

In the event of acceptance, this file will be published in conjunction with your paper and will include the anonymous referee

reports, your point-by-point response and all pertinent correspondence relating to the manuscript. If you do NOT want this file to be published, please inform the editorial office at contact@embomolmed.org.

1) a .docx formatted version of the manuscript text (including Figure legends and tables)

2) Separate figure files*

3) supplemental information as Expanded View and/or Appendix. Please carefully check the authors guidelines for formatting Expanded view and Appendix figures and tables at <https://www.embopress.org/page/journal/17574684/authorguide#expandedview>

4) a letter INCLUDING the reviewer's reports and your detailed responses to their comments (as Word file).

5) The paper explained: EMBO Molecular Medicine articles are accompanied by a summary of the articles to emphasize the major findings in the paper and their medical implications for the non-specialist reader. Please provide a draft summary of your article highlighting

This may be edited to ensure that readers understand the significance and context of the research.

Please refer to any of our published articles for an example.

6) Author contributions: the contribution of every author must be detailed in a separate section.

7) EMBO Molecular Medicine now requires a complete author checklist

(<https://www.embopress.org/page/journal/17574684/authorguide>) to be submitted with all revised manuscripts. Please use the checklist as guideline for the sort of information we need WITHIN the manuscript. The checklist should only be filled with page numbers where the information can be found. This is particularly important for animal reporting, antibody dilutions (missing) and exact values and n that should be indicated instead of a range.

8) Every published paper now includes a 'Synopsis' to further enhance discoverability. Synopses are displayed on the journal webpage and are freely accessible to all readers. They include a short stand first (maximum of 300 characters, including space) as well as 2-5 one sentence bullet points that summarise the paper. Please write the bullet points to summarise the key NEW findings. They should be designed to be complementary to the abstract - i.e. not repeat the same text. We encourage inclusion of key acronyms and quantitative information (maximum of 30 words / bullet point). Please use the passive voice. Please attach these in a separate file or send them by email, we will incorporate them accordingly.

You are also welcome to suggest a striking image or visual abstract to illustrate your article. If you do please provide a jpeg file 550 px-wide x 300-600px high.

9) A Conflict of Interest statement should be provided in the main text

10) Please note that we now mandate that all corresponding authors list an ORCID digital identifier. This takes <90 seconds to complete. We encourage all authors to supply an ORCID identifier, which will be linked to their name for unambiguous name identification.

Currently, our records indicate that the ORCID for your account is 0000-0002-9168-5628.

Link Not Available

11) Include a Reagents and Tools Table as part of the Methods section, which can be downloaded from our author guidelines (<https://www.embopress.org/page/journal/17574684/authorguide#structuredmethods>)

Photos 400-800 DPI

*Additional important information regarding figures and illustrations can be found at <https://bit.ly/EMBOPressFigurePreparationGuideline>. See also figure legend preparation guidelines: <https://www.embopress.org/page/journal/17574684/authorguide#figureformat>

***** Reviewer's comments *****

Referee #1 (Comments on Novelty/Model System for Author):

Referee #1 (Remarks for Author):

Carlini et al greatly improved the manuscript entitled „Identification of p38 MAPK inhibition as a neuroprotective strategy for combinatorial SMA therapy" which - to my opinion - can be published in its current form.

However, the authors may want to further improve consistency and readability including more data on motor neuron counts into the main figure and moving some data on central synapse counts into the supplement. The authors report that they do not see a therapeutic effect of the combinatorial treatment on *Smn2B^{-/-}* mice. They argue that the reason for that is a lack of pronounced motor neuron death. Therefore, they further argue that the effect relies on ameliorating motor neuron death while not having an effect on central synapse loss. Why then focus on central synapses in Figure 7C-D (and Fig. 8D) which cannot explain the effects seen on survival and motor function (Figure 6)? Consistent with their hypothesis there is no difference in number of synapses on MN cell bodies in mice treated with the combination compared to SMN (P8) rescued mice only (turquoise vs green bar). The authors may want to substitute Figure 7C-D central synapse counts by motor neuron counts in the same animals/treatment groups and move central synapse counts into the supplement. However, this would be a minor amelioration of an already excellent manuscript.

Response to Reviewers – EMM-2025-21654-V2, Carlini et al.**Point by point response****Reviewer #1:**

Carlini et al greatly improved the manuscript entitled „Identification of p38 MAPK inhibition as a neuroprotective strategy for combinatorial SMA therapy" which - to my opinion - can be published in its current form.

We thank the reviewer for finding our revised manuscript greatly improved and for recommending its publication in the current form.

However, the authors may want to further improve consistency and readability including more data on motor neuron counts into the main figure and moving some data on central synapse counts into the supplement. The authors report that they do not see a therapeutic effect of the combinatorial treatment on *Smn2B/-* mice. They argue that the reason for that is a lack of pronounced motor neuron death. Therefore, they further argue that the effect relies on ameliorating motor neuron death while not having an effect on central synapse loss. Why then focus on central synapses in Figure 7C-D (and Fig. 8D) which cannot explain the effects seen on survival and motor function (Figure 6)? Consistent with their hypothesis there is no difference in number of synapses on MN cell bodies in mice treated with the combination compared to SMN (P8) rescued mice only (turquoise vs green bar). The authors may want to substitute Figure 7C-D central synapse counts by motor neuron counts in the same animals/treatment groups and move central synapse counts into the supplement. However, this would be a minor amelioration of an already excellent manuscript.

We appreciate the Reviewer's suggestion. However, after careful consideration, we have maintained the original figure organization as agreed upon during consultation with the Editor.

We feel that presenting the analysis of both types of synapses (NMJ and proprioceptive) in Figure 7 (P11) and Figure 8 (P21) is key to document the distinct kinetics of synaptic rewiring and to support the mechanistic interpretation for the synergistic therapeutic effects provided by MW150-dependent neuroprotection and SMNC3-mediated synaptic rewiring.

As for adding motor neuron counts in Figure 7 (P11 time point), please note that these data are already shown in Figure 4F-G and 5-G-H. The analysis of motor neuron counts at P21 was added during manuscript revision and is presented in Figure EV4.

Lastly, the *Smn2B/-* mice were not subjected to any combinatorial treatment because they do not contain the SMN2 genes and, therefore, are not amenable to treatment with SMN inducing drugs, which were only used in the SMN Δ 7 model. A clarification has been added to the manuscript text.

20th Aug 2025

Dear Prof. Pellizzoni,

We are pleased to inform you that your manuscript is accepted for publication and is now being sent to our publisher to be included in the next available issue of EMBO Molecular Medicine.

Zeljko Durdevic
Senior Editor
EMBO Molecular Medicine
